# SiR-actin-labelled granules in Foraminifera: Patterns, dynamics, and hypotheses

Jan Goleń[1], Jarosław Tyszka[1], Ulf Bickmeyer[2], Jelle Bijma[2]

[1]ING PAN – Institute of Geological Sciences, Polish Academy of Sciences, Research Centre in Cracow, Biogeosystem Modelling Group, Senacka 1, 31-002 Kraków, Poland
[2]AWI – Alfred-Wegener-Institut Helmholtz-Zentrum für Polar- und Meeresforschung, Am Handelshafen 12, 27570 Bremerhaven, Germany

*Correspondence to:* Jan Goleń (ndgolen@cyf-kr.edu.pl)

**Abstract.** Recent advances in fluorescence imaging facilitate actualistic studies on organisms used for palaeoceanographic reconstructions. Observations of cytoskeleton organisation and dynamics in living foraminifera foster understanding of morphogenetic and biomineralisation principles. This paper describes the organisation of a foraminiferal actin cytoskeleton using *in vivo* staining based on fluorescent SiR-actin. Surprisingly, the most distinctive pattern of SiR-actin staining in Foraminifera is the prevalence of SiR-actin-labelled granules (ALGs) within pseudopodial structures. Fluorescent signals obtained from granules dominate over dispersed signals from the actin meshwork. SiR-actin-labelled granules are small (around 1 μm in diameter) actin-richstructures, demonstrating a wide range of motility behaviours from almost stationary oscillating around certain points to exhibiting rapid motion. These labelled microstructures are present both in Globothalamea (*Amphistegina*, *Ammonia*) and Tubothalamea (*Quinqueloculina)*. They are found to be active in all kinds of pseudopodial ectoplasmic structures, including granuloreticulopodia, globopodia, and lamellipodia, as well as within the endoplasm. Several hypotheses are set up to explain either specific or nonspecific actin staining. Two hypotheses regarding their function are proposed, if specific actin labelling is taken into account: (1) Granules are involved in endocytosis and intracellular transport of different kinds of cargo; (2) They transport prefabricated and/or recycled actin fibres to the sites where they are needed. These hypotheses are not mutually exclusive. The first hypothesis is based on the presence of similar actin structures in fungi, fungi-like protists and some plant cells. The later hypothesis is based on the assumption that actin granules are analogous to tubulin paracrystals responsible for efficient transport of tubulin. Actin patches transported in that manner are most likely involved in maintaining shape, rapid reorganisation, and elasticity of pseudopodial structures, as well as in adhesion to the substrate. Finally, our comparative studies suggest that a large proportion of SiR-actin-labelled granules probably represent fibrillar vesicles and elliptical fuzzy coated vesicles often identified in TEM images.

## 1 Introduction

Since Foraminifera were firstly recognised by science in the beginning of 19th century, thanks to works of d'Orbigny (Lipps et al., 2011), they became subject of extensive studies. Most Foraminifera species create shells (tests) that have great potential

for preservation in the fossil record and are primarily important in Earth Science disciplines. Application of foraminiferal research includes among others: biostratigraphy, palaeoclimatology, palaeo/environmental studies and oil and gas exploration. As a consequence, morphology, geochemical composition and evolution of their tests are much better understood than their biology. However, to properly understand fossils, it is essential to take into account the physiology of the living organisms.

Recognition of this problem together with advances in research methods has led to an increasing number of studies concerning ultrastructure of foraminiferal cytoplasm and its role in biomineralisation (e.g. Spero 1988; de Nooijer et al., 2009; Tyszka et al., 2019).

Cytoplasm in Foraminifera can be divided into two parts: ectoplasm (outside the test) and endoplasm (inside the test) (e.g., Boltovskoy and Wright, 1976). They differ not only in location relative to the test, but also in composition and appearance

under the light microscope: endoplasm is much thicker and usually is coloured even in the non-symbiotic species, ectoplasm is less dense and transparent. In addition, many organelles such as nuclei, ribosomes, Golgi apparatus are reported to occur only in the endoplasm (Bowser and Travis, 1991). The most prominent ectoplasmic structures in Foraminifera are pseudopods, which have a characteristic granular appearance, distinguishing Foraminifera from amoeba such as *Gromia* (Cavalier-Smith et al., 2018). This versatile network of branching pseudopods is involved in motility (Kitazato, 1988), feeding, construction of

the test and responding to environmental stimuli (Goldstein, 1999). As granuloreticulopodia are typically the outermost part of foraminiferal cell these structures must fulfil a crucial role in that process. The presence of granuloreticulopodia is the most fundamental morphological feature of Foraminifera and it must have appeared very early in the evolutionary history of this group (Pawlowski et al., 2003). Foraminifera probably owe much of their evolutionary success to this versatile structure.

Despite numerous studies concerning structure and function of granuloreticulopodia, many aspects of their organisation and

physiology are still unclear. The most striking reticulopodial features are fine granules that exhibit various behaviours. Granules are moving rapidly along threads of pseudopods and even along a single thread they exhibit movement in both directions (Jahn and Rinaldi, 1959; Kitazato, 1988). There are numerous different categories of granules including food particles (phagosomes), defecation vacuoles, mitochondria, dense bodies, clathrin-coated vesicles, elliptical vesicles (Bowser and Travis, 1991). Granuloreticulopodia are not the only forms of exoplasmic (pseudopodial) structures present in

Foraminifera. Pseudopodial structures are also represented by lamellipodia (*sensu* Travis et al., 1983; Tyszka et al., 2019), globopodia and frothy pseudopodia (*sensu* Tyszka et al., 2019). All these pseudopodial structures are highly functional that is well expressed by their different morphologies and temporal organisation linked to life strategies and behaviour.

Previous studies have shown that pseudopodial structures in Foraminifera depend on cytoskeleton organisation that includes microtubules (built from tubulin proteins) and actin filaments (Travis et al., 1983; Koonce et al., 1986b; Tyszka et al., 2019).

Latest investigations on morphogenesis of foraminiferal shells revealed that chamber formation and biomineralisation are directly supported by actin meshworks and closely associated with microtubular networks (Tyszka et al., 2019). The same study also reported granularity of actin detected under fluorescent light of live actin stained foraminifera. This active, bi-directional granular organisation of actin was observed in all types of pseudopodial structures, including reticulopodia, as well as globopodia and lamellipodia during chamber formation of *A. lessonii* d'Orbigny. Motile granules followed relatively straight

and often anastomosing tracks (Tyszka et al., 2019, movies S1-S6). However, the authors neither focused on this aspect of actin organisation nor on its dynamics. Structural and functional relationships between actin meshworks and their association with actin granularity have never been described nor interpreted (see Frontalini et al., 2019).

This paper is an attempt to fill the gap in our knowledge on actin organisation and dynamics in Foraminifera. Therefore, the main objectives of this study include:

(a) live fluorescent labelling of actin within ectoplasmic (pseudopodial) structures during various behavioural and/or physiological activities;

(b) live fluorescent co-labelling of mitochondria to identify a relative localisation and dynamics of granules represented by mitochondria and SiR-actin-labelled structures;

(c) identification and detailed description of the actin cytoskeleton organisation in Foraminifera with particular focus on its granularity and dynamics by means of live fluorescence imaging;

(d) assessment of unspecific labelling risk in order to evaluate reliability of staining results;

(e) comparative analysis of published images of cytoplasmic foraminiferal ultrastructure observed in Transmission Electron Microscope (TEM), to identify granular structures on TEM images that may correspond to SiR-actin labelled granules;

(f) interpretation and discussion of working hypotheses regarding the functionality of actin granularity and its evolutionary consequences. This will take into account the physiological role of similar actin structures identified and described so far in other organisms.

## 2 Materials and Methods

### 2.1 Foraminiferal culture

Experiments were performed on 3 species of foraminifera *Amphistegina lessonii* d'Orbigny, *Ammonia* sp., *Quinqueloculina* sp. They belong to both main classes of multilocular foraminifera, i.e. first two species belong to Globothalamea and the third one to Tubothalamea. Specimens of *A. lessonii* were collected from the coral aquarium of Burgers' Zoo in Arnhem (the Netherlands). This aquarium contains a diverse assemblage of corals and other organisms from the Indo-Pacific, among them there are around fifty species of benthic foraminifera, including *A. lessonii* (Ernst et al., 2011). Samples of sediment with living foraminifera were transferred to the Alfred Wegener Institute (AWI) in Bremerhaven (Germany) and the Institute of Geological Sciences of the Polish Academy of Sciences (ING PAN) in Kraków (Poland), where cultures were established in 10 l aquaria immediately after delivery. Samples containing *Quiqueloculina* sp. were collected in the oceanarium as a part of the Africarium in the Zoo Wrocław (Poland) and transported to the ING PAN in Kraków, where they were cultured in 50 l aquaria. Cultures of *A. lessonii* were kept in 12:12 light:dark cycles and natural sea water (salinity of 34). Samples of mud with *Ammonia* sp.

were collected from tidal flats in Dorum (Lower Saxony, Germany), transported to ING PAN (Kraków) and stored in 0.25-0.5 l bottles with natural sea water (salinity of 34) in thermostatic cabinet (12:12 light:dark cycle; 8 °C).

We employed two different methods of sample preparation for observation during experiments. (1) Searching for asexual reproduction events, we monitored large individuals of *A. lessonii* climbing the glass walls of the aquaria. The juvenile individuals (2-5 days old) were picked using a fine paint brush and transferred into a sterile imaging Petri dish (ibidi® polymer coverslip bottom) containing 2 ml of clean culture medium (up to ten individuals per dish). After one dark phase of the light:dark cycle, when individuals attached to the coverslip bottom, they were examined under a binocular looking for pseudopodial activity and chamber formation. This method was applied to run chamber formation experiments (for more details see supplement to Tyszka et al. 2019).

(2) In the second method adult individuals of *A. lessonii, Ammonia* sp. and *Quiqueloculina* sp. were picked from the culture aquaria or bottles and cleaned with fine paint brushes under the binocular to remove algae and grains of sediment covering the specimens. Then, they were transferred to glass bottom Petri dishes previously treated with hydrochloric acid over 16 hours containing 2 ml low calcium artificial sea water prepared as described in Bowser and Travis (2000). The reason for using low calcium artificial sea water was to avoid so called beading response (Bowser and Travis, 2000). This response causes transition of pseudopodia into droplets and may be provoked by different chemical and physical factors (Travis et al., 1983). After acclimation to the low calcium sea water, when reticulopodia were extended and adhered to the glass bottom, specimens were stained and observed. Toyofuku et al., (2008) reported that incubation of foraminifera in low calcium sea water for 3 day significantly decreases level of intracellular calcium, in result the presence of intracellular calcium cannot be deducted with fluorescence probes. However, we always started our experiments immediately after reticulopodia were extended (usually within 0.5 h after the transfer to low calcium sea water) and within this time the signal did not decrease below the detectable level. The second method was employed for observations of reticulopodia.

## 2.2 Fluorescent probes and staining procedure

We focused on staining F-actin with SiR-actin but also used Mitotracker Green to stain mitochondria and calcein Red- Orange AM for staining cytoplasm. For experiments focusing on actin organisation during chamber formation (Figs. 3-4; Figs. S1-S2 and Movie S2 in Supplement) we added stock solution of probes prepared according to manufacturers' instruction directly to the imaging Petri dish with living specimens of *A. lessonii* to a final concentration of 1 μM. For experiments regarding raticulopodia (Figs. 1-2, 5-6; Figs. S3-S5 and movie S1 in Supplement) the concentration of SiR-actin was 0.5 μM and of Mitotracker Green was 1 μM. SiR-actin is a cell-permeable, fluorogenic probe labelling F-actin, thus it is suitable for live-staining (Lukinavičius et al., 2014).

After 15-20 minutes the signal was sufficient to perform observations. Calcein Red-Orange AM is a cell-permeable dye that stains the cytoplasm of living cells and is often used to indicate the viability of cells (Frontalini et al., 2019). Calcein AM is hydrolyzed in the cytosol and fluoresce in the presence of calcium ions. It differs in a chemical structure and fluorescent from

the calcein AM used for staining cytoplasm be Ohno et al. (2017). In our experiments the main purpose of using Calcein Red-Orange AM was to indicate the limits of the cytoplasm and to highlight 3D structure of pseudopodia (e.g., globopodium). Live cells structures are stained with this dye even, if they are surrounded by calcium-free artificial sea water, as calcium ions are always present within living cells. Table S1 (Supplement) summarises information on fluorescent probes used in presented research.

Absorption and emission parameters of the SiR-actin probe are overlapping with the autofluorescence of chlorophyll from endosymbionts, thus, distinguishing between SiR-actin signal and autofluorescence emitted from the endoplasm of *A. lessonii* or other species hosting endosymbionts is difficult. To minimise the problems caused by autofluorescence we focused primarily on pseudopods where endosymbiotic algae are not present. Chlorophyll and other substances present in food particles captured by pseudopods may also give some fluorescent signal. For ensuring that such food particles are absent within observed pseudopods all specimens were starved for 24 hours prior the staining and observations.

## 2.3 Fluorescence and transmitted light microscopy

Images were obtained with a Leica SP5 inverted confocal microscope at the AWI and with a Zeiss Axio Observer Z.1. equipped with ApoTome.2. at the ING PAN in Cracow. ApoTome.2. is a device enabling removal of scattered light in fluorescence imaging. It takes between 3 to 15 images with different positions of a grid placed in the light path between fluorescent lamp and the sample. On the basis of those images, the dedicated ApoTome software calculates optical sections of the sample using a structured illumination principle to enhance signal/noise ratio of the image (Weigel, 2009). In case of living samples containing moving structures it may result in multiplication of some rapidly moving objects. Because foraminiferal ectoplasmic structures are highly dynamic, we choose to set up ApoTome.2. to take only three pictures per frame and use maximum light intensity to decrease exposure time. Despite of this, the most rapidly moving objects may appear in triplicate in some images. Hardware settings used for obtaining images are summarised in Tables S2 (exposure time, binning mode, objectives and Apotome mode in experiments conducted on Zeiss Axio Observer Z.1.), S3 (hardware settings in experiments conducted on Leica SP5 inverted confocal microscope), and S4 (filter sets used for different fluorescent dyes for experiments performed on Zeiss Axio Observer Z.1.). We optimised these settings using a trial and error method. To provide additional information over all structures of observed individuals, we captured bright field images for experiments with the Leica SP5. and Nomarski contrast imaging for the experiments with the Zeiss Axio Observer Z.1. Nomarski contrast or Differential interference contrast (DIC) is a microscopy technique utilising interferometry principle for improving contrast in transparent objects (Lang, 1968).

## 2.4 Measurement of velocity of the Actin Labelled Granules

For measuring the velocities of granules in pseudopodia we used time lapse records of pseudopodia labelled with SiR-actin using the Zeiss Axio Observer Z.1. Time lapse movies consisting of 50 frames were recorded (0.419 s time interval). SiR-

actin-labelled granules (ALGs) can display very rapid movement, hence tracking requires dense time lapses. To minimise time necessary for capturing single frame we used only one fluorescent channel (without ApoTome).

TrackMate plug-in (Tinevez et al., 2017) applied to the Fiji software (Schindelin et al., 2012) was used for calculating the velocities of the ALGs. Calculating velocities required two main steps: (1) annotation of spots representing ALGs in each of the frames of the time lapse and (2) creating links between particular spots in subsequent frames in time lapse (tracks of movement of spots). The software allows for choosing several options for both of the main steps. This can be done either manually or automatically (with several different options in the latter case). We used the LoG (Laplacian of Gaussian) detector for automatic annotation. As the approximate size of the ALGs is 1 µm, we used 1 µm as a 'blob' size parameter in the LoG detector. The threshold value was set up to 100. 'Blob' is a technical term used in the tracking of any objects in during time lapses measurements. These values were optimised by trial and error to minimise two types of errors: (1) lack of annotation of some objects that can be clearly identified on the images (2) annotation as spots areas with no apparent ALGs. It is not always easy to track which blobs in frame n+1 match with those in frame n. ALGs may temporally get so close to each other that limits their separation in subsequent frames. To recognise this we allowed for merging and splitting the tracks. After testing automatic and manual options for the second step (creating links between spots), we decided to perform it manually, as automatic methods seem to give random results for the ALGs. When spots are annotated and links between spots in the subsequent frames are created velocities of spots are measured automatically.

## 3 Results

### 3.1 Identification of SiR-actin-labelled structures by fluorescence microscopy

Fluorescent SiR-actin labelling has revealed three considerably different patterns of staining in *Ammonia* sp, i.e. (1) weak but non-uniform staining following all membranous surfaces of pseudopodial structures, (2) linear or ring-like structures showing intense fluorescence, and (3) small but strongly labelled granular structures that often exhibit very rapid dynamics (Fig. 1; Movie S1 in Supplement). The term SiR-actin-labelled granules (ALGs) is introduced here for these small oval objects. Their size has been estimated to be approximately 1 µm. This is consistent with measurement of size of objects corresponding to ALGs seen in Nomarski contrast (DIC) images (Figs. 1, 2).

ALGs are present within: lamellipodia covering the foraminiferal tests in *A. lessonii* (Fig. 3; Figs. S1-S2 in Supplement) or any other structure they are attached to, finger-like rhizopodial structures, constructing outer protective envelopes of chamber formation sites (Fig. 4), reticulopodia during feeding and locomotion (Fig. S3 in Supplement). They can be identified in endoplasmic structures within the chambers of non-symbiont-bearing species such as *Quinqueloculina* sp., close to surfaces of internal walls of the test (Fig. 5). At first glance ALGs seem to show fast and random movements but actually they can display different 'behaviours'.

Structures labelled with SiR-actin can be matched with pseudopodial structures and granular microstructures identified in Nomarski contrast (DIC) or in brightfield image. Figures 1-2 present a lamellipodial structure attached to the glass surface with a weak, dispersed fluorescent signal of SiR-actin staining the F-actin meshwork. Very fine bright spots represent ALGs that match with granules observed with DIC optics.

## 3.2 Testing the selectivity of labelling of granules: SiR-actin-labelled granules vs mitochondria

Direct comparative analysis of fluorescence vs DIC images of *A. lessonii* indicate that SiR-actin-labelled granules do not overlap with all granules observed in DIC (Fig. S3 in Supplement). It means that SiR-actin does not stain all the granules observed. Therefore, labelling of ectoplasmic granules is selective. In order to test ALGs relationships with selected, well-defined granules, mitochondria were chosen for a double labelling experiments. Mitochondria were the best candidates because they had frequently been recognised within the cytoplasm, including reticulopodia (e.g., Travis and Bowser, 1986; Hottinger, 2006; Nomaki et al., 2016; LeKieffre et al., 2018a). Mitochondria usually appear oval or kidney shaped in cross section with a length in the range of 0.5 to 1 μm, although they are sometimes larger and take various, even tubular shapes (LeKieffre et al., 2018a).

MitoTracker Green has been applied in living specimens of *A. lessonii* following the procedure described above (Material and Methods). This probe selectively accumulates in the mitochondrial matrix by covalent binding to mitochondrial proteins (Presley et al., 2003). Results of replicated live experiments do not show co-localisation of ALGs and mitochondria stained by MitoTracker Green (Fig. 3; Fig. S3 and Movie S2 in Supplement). Therefore, they indicate that mitochondria and SiR-actin labelled granules are two non-overlapping categories.

## 3.3 Dynamics of SiR-actin-labelled granules

The dynamics of the ALGs (velocity and overall pattern of movement) are described separately in granuloreticulopodia and in a globopodium during chamber formation. The dynamics may vary for different locations in the cell. Not all of the ALGs have the same pattern of movement. At first glance their movement may appear chaotic, but closer analysis reveals some general patterns.

For the sake of simplicity, particular threads of granuloreticulopodia may be considered as one-dimensional structures that constrain the possible directions of the movement: they can move along the thread of reticulopodia either inward or outward. Indeed bidirectional movement along a single thread is commonly observed in *A. lessonii* (Figs. S4-S5 in Supplement), however in case of thick pseudopodial treads there may by a spatial separation: in the core of pseudopodium ALGs moves towards the cell body, while in the cortex they travel in the opposite direction (Movie S3 in Supplement). Usually one direction is dominant: when reticulopodia are formed, outward (centrifugal) transport is more common: during retraction of reticulopodia inward (centripetal) movement is prevalent. During extension of a newly formed very fine thread of pseudopodium, there usually is a single ALG at the tip of this tread (Fig. 6). Sometimes clusters of granules moving together with the same speed along a pseudopodium may be identified. As the granuloreticulopodia themselves are very dynamic

structures, it is not always possible to measure displacement of ALGs due to the absence of a stationary reference frame. Another problem is that ALGs can be so abundant in reticulopodia that they may be extremely difficult to track. To overcome this problem intervals between subsequent frames in time lapse movies were minimised. Using this strategy we recorded time lapse movies (Movie S4 in Supplement) showing a wide range of velocities of ALGs in reticulopodia of *A. lessonii* up to 15.4

μm/s. (Fig. S8 and Tables S5- S6 in Supplement).

Lamellipodia covering the test form two-dimensional sheets, resulting in a more complex pattern of displacement of ALGs than the one observed in granuloreticulopodia. There are areas dominated by directional protoplasm streaming and areas showing less organised behaviour. Accordingly, actin granules can be divided into several categories based on dominant movement patterns: (1) stationary or almost stationary ALGs that oscillate within a very narrow space; (2) ALGs showing

saltatory movements as described in Travis and Bowser (1991); (3) ALGs exhibiting extremely rapid movement that can be observed for up to a few seconds. Moreover, in some areas actin granules may move along a single line but with different velocities and in different directions. They may pass some stationary granules with no significant interaction observed.

## 4 Interpretation and Discussion

### 4.1 Assessment of unspecific fluorescent labelling risk

All microscopy techniques are associated with a risk of capturing artefacts instead of imaging target structures. In case of fluorescence microscopy the greatest danger is unspecific labelling or autofluorescence. Comparison stained individual of *A. lessonii* to unstained control shows that SiR-actin fluorescent probe indeed stains endoplasmic structures in foraminifera (Fig. S6 in Supplement). This may be caused when the concentration of the probe is too high or when the excitation intensities (or emission measurement sensitivity) are too high. Another problem might be that the probe is not specific enough and binds to

other chemical compounds in the cell, which structure mimics the target structure. Most fluorescent probes were developed and tested to study mammalian cells, therefore, the risk of unspecific fluorescent labelling should be addressed to avoid confusion and over-interpretation of the results. Foraminifera are placed in the actin phylogenetic tree with Bikonta (Flakowski et al. 2005), thus the amino acid sequence of actin in foraminifera is significantly distinct from the actin sequence in Metazoa or fungi (belonging to opisthokonts) that are subject of most intensive research on actin physiology. Fluorescent probes may

therefore interact differently with actin in Foraminifera. Moreover, Foraminifera may contain other organic compounds that mimic of actin nanostructures and therefore interact with fluorescence probes, such as SiR-actin. It should be noted, however, that our results are reproducible.

As the granular pattern of SiR-actin staining is unusual compared to other eukaryotes, it requires an extensive discussion of all possible scenarios (see Fig. S7). There are three possible scenarios in which ALGs may represent real F-actin-containing

structures that are labelled by SiR-actin probe (Fig. S6A-C in Supplement), and three additional possibilities that would reveal the observed patterns as artefacts (Fig. S6D-F in Supplement). The first and most likely scenario (Fig. S6A in Supplement) assumes that foraminifera possess granular structures filled with densely packed actin filaments that are specifically stained

with SiR-actin. These structures possibly correspond to Fibrillar Vesicles known from TEM ultrastructure studies (see below in Section 4.5.1). According to the second scenario, labelled actin filaments surround some membranous vesicles (Fig. 1B in Supplement). These vesicles are possibly involved in transport and endocytosis and F-actin probably plays role in those processes. Alternatively, they may correspond to elliptical fussy-coated vesicles described by Koonce et al. (1986) regulating

motility of reticulopodia (see below section 4.5.2 Elliptical fuzzy-coated vesicle). The third scenario assumes that actin filaments are located both inside and outside of some membrane-bound vesicles (Fig. S6C). Alternatively, the observed staining pattern may be explained as an artefact, if SiR-actin binds to another, unidentified, organic molecule that is different from, and not associated with F-actin, either inside (Fig. S6D in Supplement) or outside (Fig.S6E in Supplement) of membranous vesicles. Lastly, SiR-actin may induce assemblage of actin filaments in the areas rich in G-actin (Fig. S6F in

Supplement) as suggested by Melak et al. (2017).

The first argument supporting the reliability of SiR-actin live staining is the fact that attachment sites of pseudopodia to the substratum often demonstrate a strong fluorescence signal (Fig. 1) as predicted from the essential role of actin for adhesion (Bowser et al., 1988). Secondly, as mentioned above, granular actin structures are visible on images of fixed reticulopodia stained with phalloidin (see Koonce et al., 1986a, fig. 3C; 1986b fig. 1F). It cannot be excluded, however, that ALGs are

associated to a defence strategy to remove and dispose toxic compounds introduced into the cell. If this were true, we would expect vesicles containing those probes to be transported outward. As ALGs are often moving bi-directionally (both in- and outwards) (Figs. S4, S5 in Supplement), this hypothesis is not very convincing. In fact, ALGs' inward movement is observed when a pseudopodial structure is being withdrawn and seems to indicate relocation of labelled actin into the endoplasm. Such observations support the notion that live staining using SiR-actin is specifically labelling actin and that the inward movement

of ALGs is a functional response during withdrawal of pseudopodial structures.

Another issue that needs to be considered is the risk of interference of a probe with the physiology of actin itself as it may, for instance, cause an artificial polymerisation of F-actin (Melak et al., 2017). In that case we would expect negative interference of SiR-actin on morphogenesis and biomineralisation of new chambers. Nevertheless, such staining artefacts have never been observed (Tyszka et al., 2019). Moreover, if SiR-actin causes polymerisation of F-actin, live actin staining should have a

visible impact on organisation and motility of pseudopodia. In our observations we did not recognise any apparent long-term differences neither in morphology or nor in dynamics of reticulopodia after staining. Occasionally, we observed a temporary retraction of pseudopods immediately after adding the staining solution to the petri dish. However, after 10-15 minutes incubation with SiR-actin, this effect was not visible any more, and reticulopodia were spread out again, closely resembling the pre-staining structure and dynamics. *A. lessonii* is less prone to retraction of pseudopods and recovers faster than *Ammonia*

sp. or *Quinqueloculina* sp. after applying staining solution.

**4.2 Previous studies on actin in Foraminifera using fluorescent labelling**

The most commonly used method of fluorescent labelling of the actin cytoskeleton is phalloidin staining (Melak et al. 2017). Its utility is limited mostly to staining fixed cells. Phalloidin staining was previously employed to study the actin cytoskeleton

in reticulopodia of a few species of foraminifera, i.e. mainly *Reticulomyxa filosa* (Koonce et al., 1986a, Koonce et al., 1986b) and *Allogromia* sp. (Bowser et al., 1988). Actin staining of *R. filosa* showed cable-like structures concentrated in the cortex of the reticulopodia as a dominant pattern of actin organisation in reticulopodia. Along those structures there are visible granular actin structures in the figures of the cited publications that are not discussed or mentioned by the authors (Koonce et al., 1986a,

fig. 3C; Koonce et al., 1986b, fig. 1F). In *Allogromia* sp. the actin cytoskeleton has a different organisation depending on the location in the reticulopodium: in proximal parts of pseudopodia a thick linear fibre; in more distal regions flattened on the glass it is visible only in a few locations, resembling the SiR-actin-labelled granules in our study; in the most peripheral areas actin staining is absent (Bowser et al., 1988, fig. 1C, 2C, 3C). We suspect that the structure in the distal regions flattened on the glass correspond to the ALGs described in our paper.

Although Figure 1 demonstrates an SiR-actin-labelled linear structure and Figure 4d presents indistinct SiR-actin-labelled fibres, clear cable-like structures are absent in our study in comparison to previous publications which may be a result of different staining procedures. This is due to the fact that every probe may have affinity to different epitopes of F-actin, therefore, may not label all different F-actin-containing structures equally. The effectiveness of staining F-actin using different probes was compared by Lemieux et al. (2014). They reported that different probes did not stain all of subsets of F-actin

equally. Apparently, the staining effectiveness of F actin depends on the location of actin filaments within the cell. Even though this analysis does not include SiR-actin, the same issue may apply to this probe. The interaction between probes and F-actin may also lead to stabilisation or enhanced polymerisation of F-actin due to its structural similarities to Jasplakinolide (Melak et al., 2017). In addition, cell fixation procedures may stabilise dynamic structures or create some artefacts.

Previous studies on the dynamics of granules in Foraminifera were conducted mostly on *Allogromia* and *Astramina.* The

maximum speed of granules within reticulopodia was reported to be approximately 25 µm/s but most of them have velocities below 10 µm/s (Travis and Bowser, 1991). Velocities of ALGs fall within this range. The average speed of granules in foraminiferal pseudopodia reported by Kitazato (1988) is 13 µm/s, what is roughly comparable maximum velocity recorded by us (15.5 µm/s).

### 4.3 Main hypothesis regarding the functionality of Actin-Labelled Granules

Actin labelled granules described in this paper appear to be one of the main forms of actin cytoskeleton organisation in external cytoplasm (ectoplasm) of foraminifera. As they are ubiquitous in pseudopodia during feeding behaviour and in globopodia during chamber formation, they probably serve an important physiological role. At present, it is difficult to determine their function, however, there are a few hypotheses that could be proposed based on two sources of data.

As mentioned above there are two possible explanation of their role: (1) ALGs mediate transportation of various types of

cargo; (2) ALGs are involved in transport of prefabricated or recycled actin fibres. The following paragraphs are dedicated to the discussion of these hypotheses. Firstly, we will discuss the relation of actin granules in foraminifera to similar structures described in other organisms. There are actin patches known from some fungi and fungi-like protists. Secondly, we compare actin granules to different ultrastructures known mostly from TEM images of foraminifera. We will focus on organelles or

structures whose functions are questionable e.g. fibrillar vesicles (LeKieffre et al., 2018a; Goldstein and Richardson, 2018), and elliptical fuzzy-coated vesicles also called Motility Organising Vesicles (Travis and Bowser, 1991).

### 4.3.1 Comparison of actin structures in other organisms

Structures similar to SiR-actin-labelled granules described in Foraminifera have been found in other organisms. They are
present in water moulds: *Saprolegnia ferax* (Geitmann and Emons, 2000), *Phytophthora infestans* (Meijer et al., 2014), as well as in yeast *Saccharomyces cerevisiae* (Moseley et al., 2006; Rodal et al., 2005; Waddle et al., 1996; Winter et al., 1997), where they are abundant in buds. They are referred to as cortical actin patches in budding yeast and *S. ferax* (Geitmann and Emons, 2000) or actin plaques in *P. infestans* (Meijer et al., 2014). In those organisms they occur alongside different actin structures such as actin cables or rings.

Fluorescent images of *Saprolegnia ferax* (Geitmann and Emons, 2000) indicate that actin patches have a globular shape and diameters of approx. 0,5 μm. In yeast they appear to have a similar size. Therefore, their size is comparable to SiR-actin-labelled granules in Foraminifera. The maximum velocity of actin patches observed in yeast is 1.9 μm per second (Waddle et al., 1996), thus it is significantly lower than the velocity of actin granules in foraminifera. Cortical actin patches are most likely involved in endocytosis (Moseley et al., 2006) and cell growth (Geitmann and Emons, 2000). For instance in budding yeast
actin patches are present during budding within the daughter cell.

In foraminifera, ALGs appear in large numbers in the course of chamber formation, as well as within reticulopodia, which are known for their ability for rapid extension and retraction. Formation of a globopodium and reticulopodia in Foraminifera and budding in yeasts require quick expansion of the cytoplasm and may share similar mechanisms facilitating those processes. Assembling actin filaments may generate a physical force that can be used to provide the pressure required for expansion of
new protoplasm (Mogilner and Oster, 2003).

### 4.3.2 Comparison of SiR-actin-labelled granules to organelles identified in TEM images of foraminifera

Transmission electron microscopy (TEM) is a principal method to investigate cell ultrastructure. However, TEM images alone do not provide information on the chemical composition of certain structures. In contrast to TEM, fluorescent labelling sometimes gives detailed insight into the chemical composition of certain areas of the cell but in much lower resolution. Thus,
combining the two approaches is essential to unravel the ultrastructure and chemical make-up and thus provide clues about the function of cell components. Hence, for a better understanding of the role of actin granules in foraminiferal cells, it is important to find the corresponding structures on TEM images.

Fibrillar vesicles (FV) are the best candidates for the corresponding structures that represent ALGs under TEM. They are present in many different species of benthic foraminifera relatively abundant in various parts of their cytoplasm (Hottinger,
2006; LeKieffre et al., 2018a; Jauffrais et al., 2018; Koho et al. 2018). Their size (up to ~1 μm in diameter) and vesicular, globular shape (LeKieffre et al., 2018a; Goldstein and Richardson, 2018) correspond to ALGs (Figs. 2-3, 5-6; Figs. S3, S4-S5 in Supplement). Fibrillar vesicles appear to be separated from the cytosol by a lipid membrane (Figs. 7a, 8). Membranes

enveloping the fibrillar vesicles may not cover the entire vesicle. It may form characteristic open vase-shaped structures (Goldstein and Richardson, 2018).

Although the chemical composition of FV is uncertain we can assume from an ability to accumulate nitrogen (LeKieffre et al., 2018b) that they likely contain proteins. Internal material within FV appears to have a specific 3D net-like nanostructure. Most

fibres are oriented along the long axis of the FV, but they are not perfectly parallel. They form a network of cross-linked and branching fibres, spreading in two dominant directions and forming recurrent angles. This organisation pattern resembles the actin meshwork observed by cryoelectron tomography in *Dictyostelium* (Medalia et al., 2002) or in nano-tomography of lamellipodium in keratocyte of zebrafish (Mueller et al., 2017). The similarity in the spatial pattern of fibres inside FV to the actin meshwork leads to the conclusion that FVs contain a network of actin filaments (Fig. 7). Similar but less organised

structures of cross-linked fibres form an actin meshwork in the pseudopods of *Allogromia* (Bowser et al., 1988; Koury et al., 1985).

It is not clear how FVs are formed. LeKieffre et al. (2018a) proposed that they are produced similar to the model of forming of Golgi Vesicles published by Anderson and Lee (1991). This model assumes that they originate in the *trans* surface of Golgi apparatus, thus translation of the protein inside those vesicles must occur in the endoplasmic reticulum. This seems to be

inconsistent with our hypothesis that fibrillar material consists of prefabricated actin filaments, as actin is a cytoplasmic protein, thus its translation takes place on ribosomes in the cytosol and not in the endoplasmic reticulum (ER). However, assuming that FVs are formed by enclosing fibrillar material produced in the cytosol by the cisternae of Golgi apparatus may resolve this issue. This assumption agrees with findings by Goldstein and Richardson (2018) that the membrane may not cover the entire vesicle.

It is worthwhile to mention that Anderson and Bé (1976) described in planktic foraminifera another subcellular structure called the fibrillar system or the fibrillary bodies (according to Hemleben et al., 1989; Schiebel and Hemleben, 2017). Spero (1988) presented this system, which contained proteins involved in construction of a protective envelope during chamber formation in *Orbulina universa*. However, it is not clear, whether these structures represent structurally and functionally analogous organelles to FVs. Spero (1988, - see figs. 4e, f, 5d) documented vesicles using TEM that resemble FVs and are associated

with the "primary organic membrane" during chamber formation. In fact, "fibrillar" as a descriptive term seems to describe different filamentous structures at different spatial scales. Fibrillar vesicles show a fibrillar internal ultrastructure, in contrast to the fibrillar system that represent "massive fibrous deposits" constructed from individual tubular structures called fibrillar units (see Spero, 1988). Therefore, the fibrillar system is often tubular that contrasts to granular (vesicular) appearance of FVs and ALGs. Nevertheless, Hemleben et al. (1989) note that fibrillar bodies originate in the cytoplasm inside the test as small

vacuoles filled with densely packed fibrous material and they typically enlarge and expand as they are transferred outside the test. However, the rhizopodia of *Orbulina universa* may contain small vacuoles resembling FVs, e.g. object described as a vesicle containing adhesive substance in fig. 3.5(6) in Schiebel and Hemleben (2017). More comparative studies are needed to reveal whether FVs in benthic species are homologous to the fibrillar system in planktic ones.

Finally, Elliptical fuzzy-coated vesicles are additional ultrastructural cellular components that may correspond to ALGs. These vesicles are structures unique to Foraminifera. They include elongated structures that are typically approximately 300 nm in length identified in TEM images of reticulopodia (Koury et al., 1985; Travis and Bowser, 1991). Elliptical fuzzy-coated vesicles, consist of a membrane coated with an unknown material having a characteristic fibrillar appearance. They are reported to be involved in regulation of motility, thus, the term Motility Organising Vesicles was coined to describe those structures (Travis and Bowser, 1991). Material coating these organelles shows characteristic fuzzy appearance that might resemble actin mesh.

## 4.4 Functional implications, evolutionary consequences, and future research prospect

SiR-actin-labelled granules (ALGs) are highly dynamic structures that are abundant in foraminiferal ectoplasm (Figs. 1-3, 5-6). They are small organelles involved in the physiology of granuloreticulopodia and other types of pseudopods, some of them directly involved in morphogenesis of new chambers and biomineralisation of the wall (see Tyszka et al., 2019). As they are ubiquitous in the cells of many species of both globothalamean and tubothalamean foraminifera (sensu Pawlowski et al., 2013), they have most likely evolved very early during evolution of Foraminifera. It seems very likely that they correspond to fibrillar vesicles or fuzzy coated vesicles observed in much higher resolution using TEM (Figs. 8). More studies are needed to corroborate or refute this hypothesis, particularly applying correlative light and electron microscopy as a crucial method to solve this puzzle.

The second question that should to be addressed regards the presence of analogue structures in other eukaryotic organisms. Indeed, in some fungi or fungi-like protists similar actin structures have been identified in several previous studies (Geitmann and Emons, 2000; Meijer et al., 2014; Moseley et al., 2006; Rodal et al., 2005; Waddle et al., 1996; Winter et al., 1997). It is too early to state whether all these structures serve the same physiological function and share the same evolutionary origin. However, there are some facts suggesting that this actually may be the case. Firstly, all of them have similar size and tend to be concentrated in a cortical layer of protoplasm just under the plasma membrane. Moreover, all the cells containing them have the ability to rapidly expand the volume of protoplasm and actin networks/patches, which may be involved in generating the force needed in this process. Investigation of the molecular basis of actin cytoskeleton regulation in broad phylogenetic context is required to address this issue.

Our working hypothesis is that ALGs most likely play a crucial role in intracellular transport, that may be two-fold: (1) they may be involved in transport of various cargo inward (endocytosis) or outward (exocytosis), and/or (2) they facilitate transfer of prefabricated actin filaments from endoplasm to the external parts of the foraminiferal cell. If the second hypothesis is correct, ALGs are fundamental for extension and adhesion of reticulopodia, as well as formation and shaping the glopopodium during chamber formation.

Our hypothesis may solve the puzzle of efficient transport of proteins within extensive pseudopodial networks. In Foraminifera, ribosomes are absent in the pseudopodial cytoplasm (Travis and Bowser, 1991), and in consequence protein synthesis is restricted to the endoplasm. Therefore, foraminifera must have mechanisms to efficiently transport proteins needed for the

formation of extensive pseudopodial networks. This issue applies primarily to the transportation of the cytoskeletal proteins that are in high demand within reticulopodia due to their critical role in morphogenesis and movement of this network. Simple diffusion of monomers of tubulin and assembly of microtubules on site may not be sufficient enough (Bowser and Travis, 2002). Hence it was proposed that foraminiferal cell use tubulin paracrystals as a storage of prefabricated MT (Travis and
Bowser 1991). Here, we suggest an analogous mechanism for efficient actin transport in form of microfilaments. This mode of transport facilitates a rapid formation, restructuring, and retraction of actin meshwork.

Such functional mechanisms employed for optimisation of intracellular motility of building blocks, pseudopodial dynamics and their overall morphogenesis may be one of the main evolutionary adaptations specific to Foraminifera and possibly to related phylogenetic taxa included into the phyllum Retaria (see Cavalier-Smith et al., 2018). Similar granuloreticulopodial
organisation of pseudopods is known from Radiolaria (Anderson, 1976; Anderson, 2012). Radiolaria, also called Radiozoa are very likely a sister group of Foraminifera (Burki et al., 2010; Cavalier-Smith et al., 2018). It is not clear if all types of granules in ectoplasm of Radiozoa and Foraminifera are the same. It has been reported that granules in radiolarian pseudopodia include mitochondria, digestive and defecation vacuoles, and osmophilic granules (Anderson, 2012).

Molecular phylogeny based on conservative actin gene sequences suggests that actin in Foraminifera evolved in higher rates
than in most other eukaryotes (Keeling, 2001). Moreover, duplication of gene encoding actin has occurred early in evolution of a lineage containing Foraminifera resulting in the presence of two paralogs of that gene in many species (Flakowski et al., 2005). There is some evidence that this duplication is shared by the group Acantharea belonging to Radiolaria (Burki et al., 2010). However, in at least some Foraminifera, actin genes have been duplicated many times forming extraordinarily diverse gene families as in *Reticulomyxa filosa*. It has been suggested that the diversification of actin genes was a key step in evolution
of mechanisms of rapid transport between reticulopodia and the cell body (Glöckner et al. 2014). This suggests that physiology, dynamics, organisation and function of the actin cytoskeleton in Foraminifera may differ significantly from most other organisms. More studies are essential for the understanding of the physiological functions of the actin cytoskeleton, including:

(1) research regarding the expression of actin;

(2) identification of actin-binding proteins in Foraminifera;
(3) experiments on inhibition of actin formation during different behaviour (feeding, chamber formation, locomotion etc.);

(4) imaging of actin structures with more refining methods including correlative light- and electron microscopy or super-resolution confocal microscopy.

## 5 Conclusions

This paper presents results of live fluorescent labelling of actin in Foraminifera with a focus on ectoplasmic (pseudopodial)
structures during various behavioural and physiological activities. Fluorescence labelling has revealed three considerably different SiR-actin-labelled patterns that include: (1) weak but not uniform staining following all membranous surfaces of pseudopodial structures (Figs.1, 2), (2) linear or ring-like structures showing intense fluorescence (Fig. 1), and (3) small,

strongly labelled granular structures that often exhibit very rapid dynamics (Figs. 2-3, 5-6; Figs. S4-S5 and Movies S2-S3 in Supplement).

The granular appearance is the principal characteristic of actin cytoskeleton in all studied foraminiferal taxa. SiR-actin-labelled granules (ALGs) have been described as small (ca. 1 µm in diameter), oval and dynamic objects that are numerous in pseudopodia, but present in endoplasm as well. Besides ALGs, the actin cytoskeleton in foraminiferal pseudopodia may form a linear and ring-like structures (Fig. 1).

Co-labelling of mitochondria with Mitotracker Green and actin cytoskeleton with SiR-actin has been performed in order to verify whether ALGs overlap with mitochondria as a test for the selectivity of granules labelling. As presented images (Fig. 3; Figs. S2-S3 in Supplement) indicate, there is very little co-localisation between those two types of organelles, however, ALGs and mitochondria probably constitute the majority of granules present in pseudopodia.

A detailed interpretation of the images is given, including the risk that ALGs may be a result of unspecific labelling. Presented arguments allow to exclude this possibility. Furthermore, the relation of ALGs to similar structures found in other eukaryotes (mostly some fungi, fungi-like protists) has been discussed. It has been proposed that a main function of ALGs in physiology of Foraminifer is facilitating transportation of different types of cargo, most likely including transport of prefabricated and/or recycled actin filaments themselves. Finally, the question regarding the correspondence of ALGs to objects known from published TEM images has been addressed. According to our presented hypothesis, most of ALGs correspond to fibrillar vesicles (see LeKieffre et al., 2018a; Goldstein and Richardson, 2018) and/or elliptical fuzzy-coated vesicles (Travis and Bowser, 1991). This is still a working hypothesis that should be verified by correlative TEM-fluorescence methods.

**6 Information about the Supplement**

The Supplement contains 4 movies, 8 additional figures showing different actin structures in Foraminifera and 6 tables.

**Author contributions**

Author contributions: J.G. designed research, performed research, analysed data, cultured foraminifera, wrote the paper and prepared graphics; J.T. proposed and supervised research; U.B. and J.B. provided access and infrastructure at AWI; J.T. consulted interpretations; J.T., U.B., and J.B. corrected the text.

**Competing interests**

The authors declare that they have no conflict of interest.

**Acknowledgements**

The authors thank Joan M. Bernhard (WHOI) and Jeffrey Travis (SUNY) for valuable comments on methodological aspects of fluorescent labelling, as well as Karina Kaczmarek for help with maintenance of foraminiferal culture at the AWI, Karolina Kobos, Anna Spadło and Anna Wójtowicz for help with culture at the ING PAN. We are also grateful to Max Janse from
Burgers' Zoo in Arnhem and Jakub Kordas form Zoo Wrocław for providing us with samples of living foraminifera. The authors highly acknowledge Samuel Bowser, Takashi Toyofuku, and anonymous referee for constructive reviews that helped us improve our manuscript. JG and JT received support from the Polish National Science Centre (UMO-2015/19/B/ST10/01944).

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

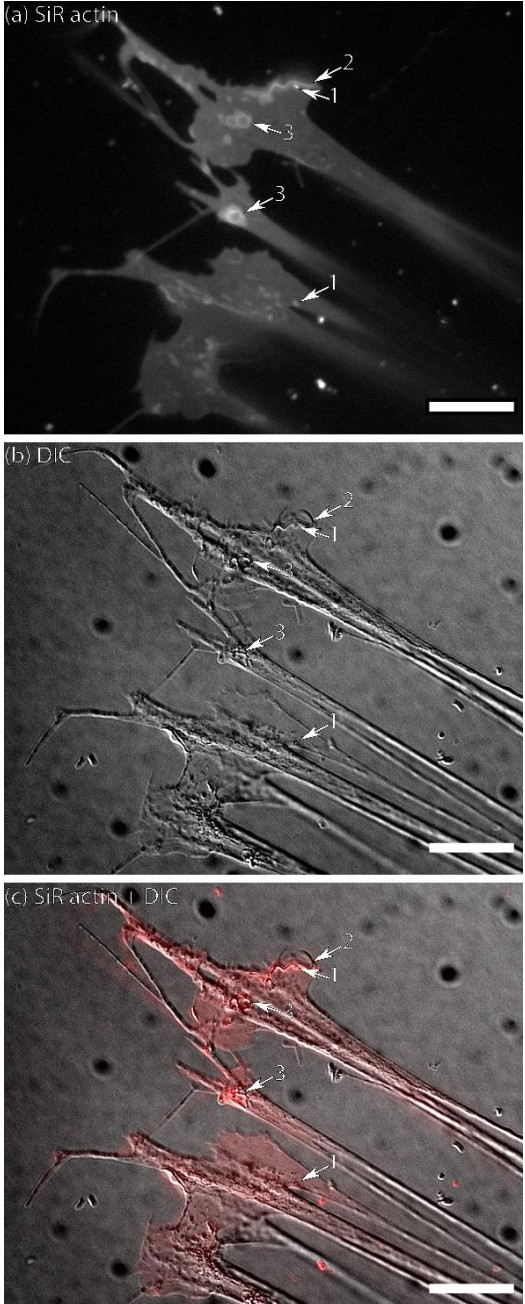

**Figure 1: Flattened lamellipodia of living *Ammonia* sp. attached to glass:** (a) fluorescence of SiR-actin-labelled structures, (b) DIC image of the same area, (c) merged image of fluorescence and DIC channels (since the reticulopodia were moving, the DIC image is slightly shifted in relation to fluorescent one). Numbers indicates: 1 – actin labelled granules (ALGs); 2 – linear SiR-actin-labelled structures; 3 – SiR-actin-labelled rings. Note weak but not uniform SiR-actin-labelling following all membranous surfaces of pseudopodial structures. The linear structure (2) was subsequently transformed into ring structure (see Movie S1 in Supplement). Structures corresponding to ALGs, SiR-actin-labelled ring and linear structures can be seen in DIC image. Conventional fluorescence images were obtained with *Zeiss Axio Observer Z.1*. Scale bar 20 μm.

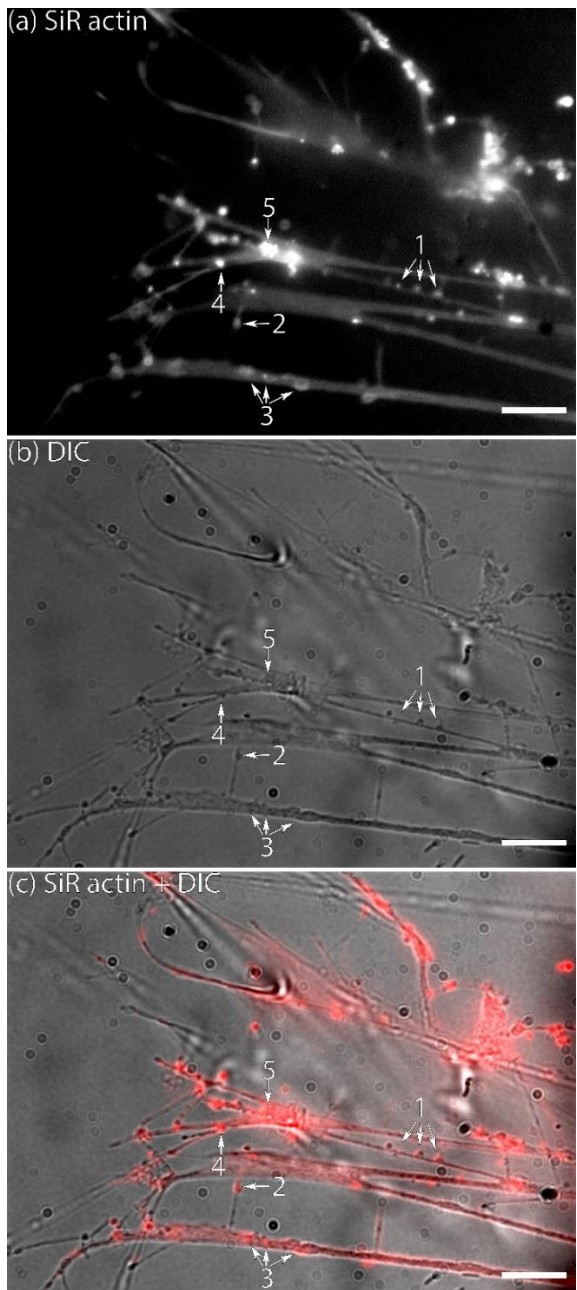

**Figure 2: Pseudopodia of living _Ammonia_ sp. attached to glass:** (a) conventional fluorescence of SiR-actin-labelled structures, (b) DIC image of the same area, (c) merged image of fluorescence and DIC channels (since the reticulopodia were moving, the DIC image is slightly shifted in relation to fluorescent one). Weak but not uniform actin-labelling following all membranes can be seen in pseudopodia. Numbers
5   indicates: 1 – group of tree SiR-actin-labelled granules (ALGs) transported along one thread of pseudopodia; 2 – actin in the tip of thin fillopodium; 3 – larger SiR-actin-labelled areas showing smudgy fluorescence weaker than in most ALGs; 4 – single ALG in bifurcation of retiulopodia; 5 – group of very bright densely packed ALGs in the thick reticulopodium. Conventional fluorescence images was obtained with _Zeiss Axio Observer Z.1_. Scale bar 10 µm.

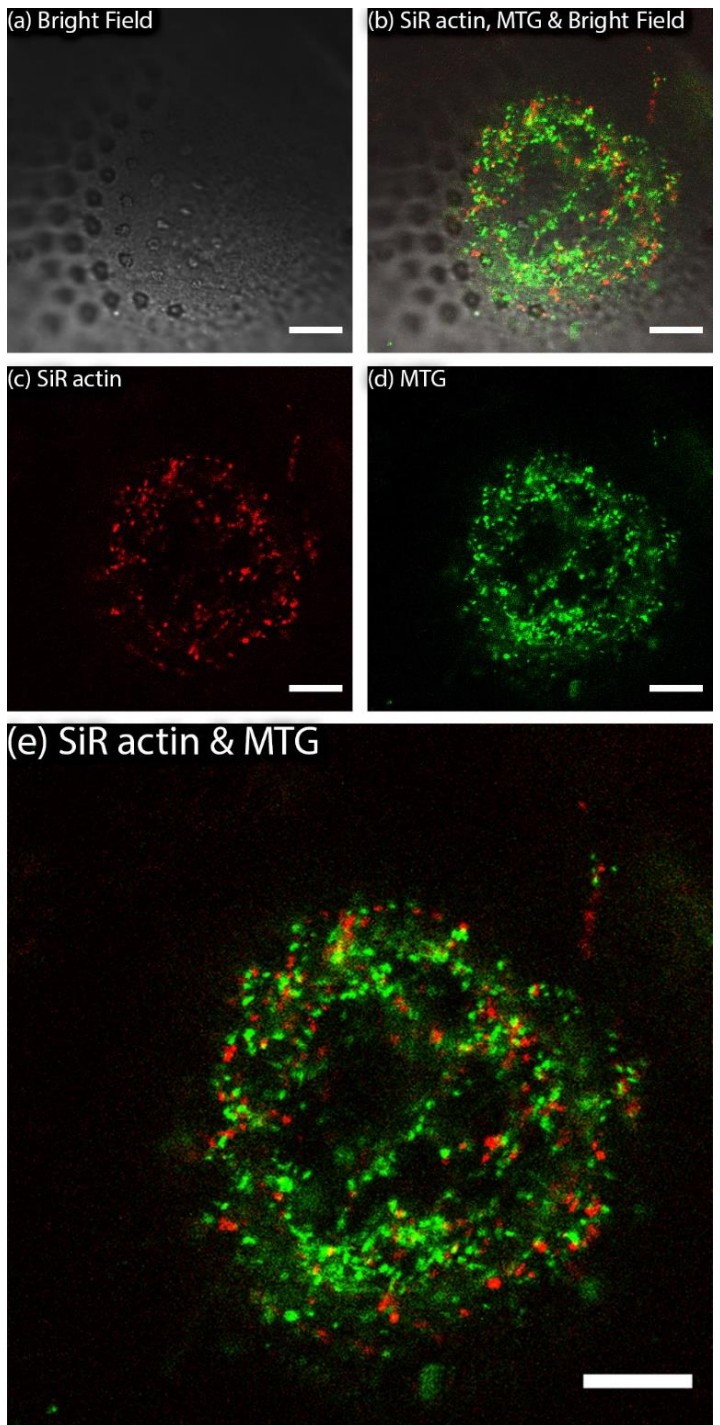

**Figure 3: SiR-actin-labelled granules (ALGs) and mitochondria in cross-section of a newly forming chamber in *Amphistegina lessonii*** during biomineralisation (pores are already visible in transmitted light). ALGs and mitochondria do not show co-localisation. Images were obtained with Leica SP5 inverted confocal microscope. For the entire time lapse see Supplementary Material 2. Scale bar 10 µm.

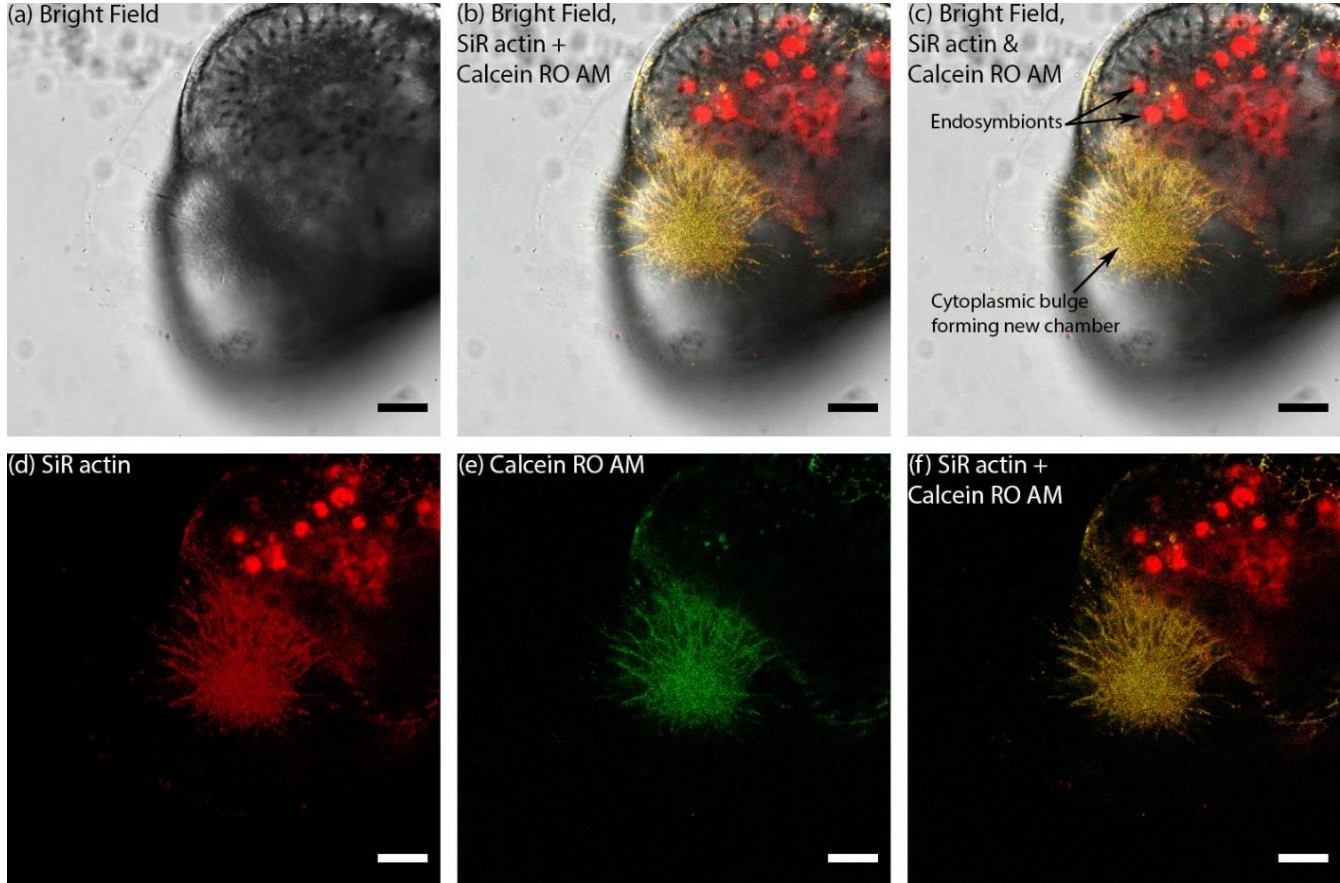

**Figure 4: Organisation of actin within in finger-like structure preceding globopodium during chamber formation in *Amphistegina lessonii*** compared with localisation of cytoplasm stained with calcein red orange AM. Images was obtained with *Leica SP5* inverted confocal microscope. Scale bar 20 μm.

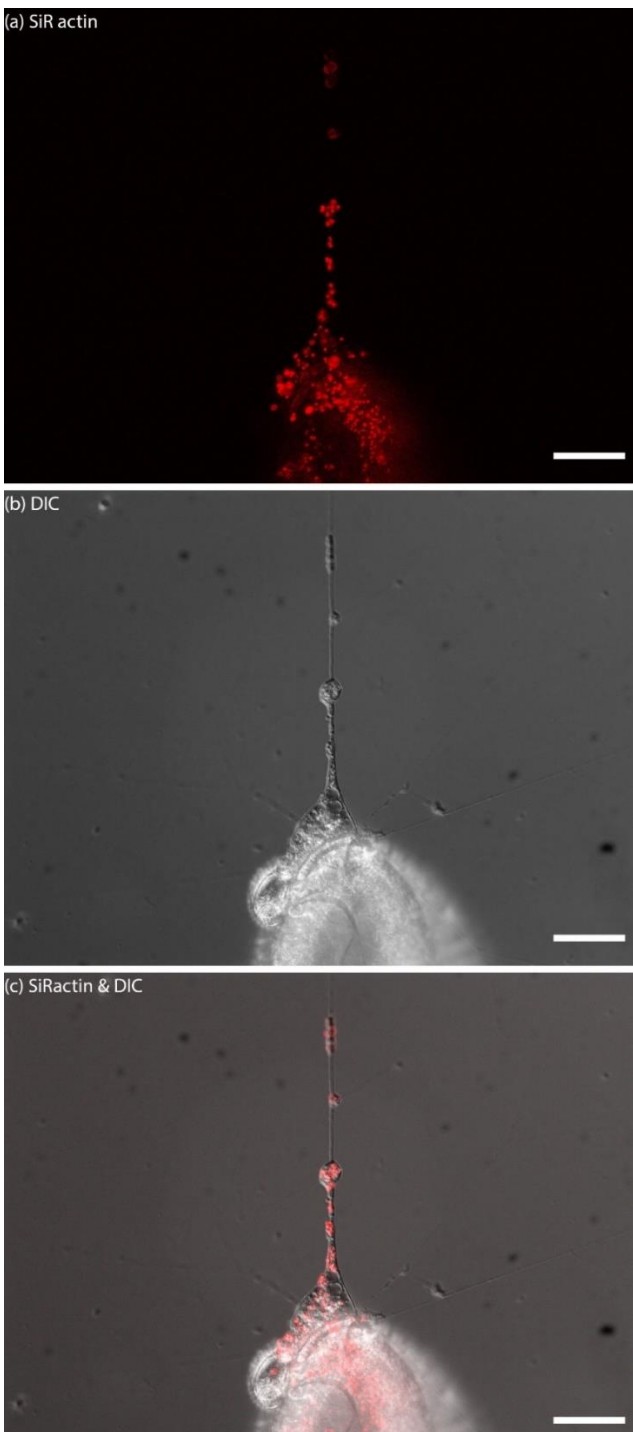

**Figure 5: SiR-actin-labelled granules (ALGs) in pseudopodia and endoplasm of *Quinqueloculina* sp.** Top image presents actin stained with SiR-actin, middle image presents images obtained with DIC optics (inverted LUT), bottom column presents merged fluorescent and DIC channels. Conventional fluorescence images obtained with *Zeiss Axio Observer Z.1*. Scale bar 50 μm.

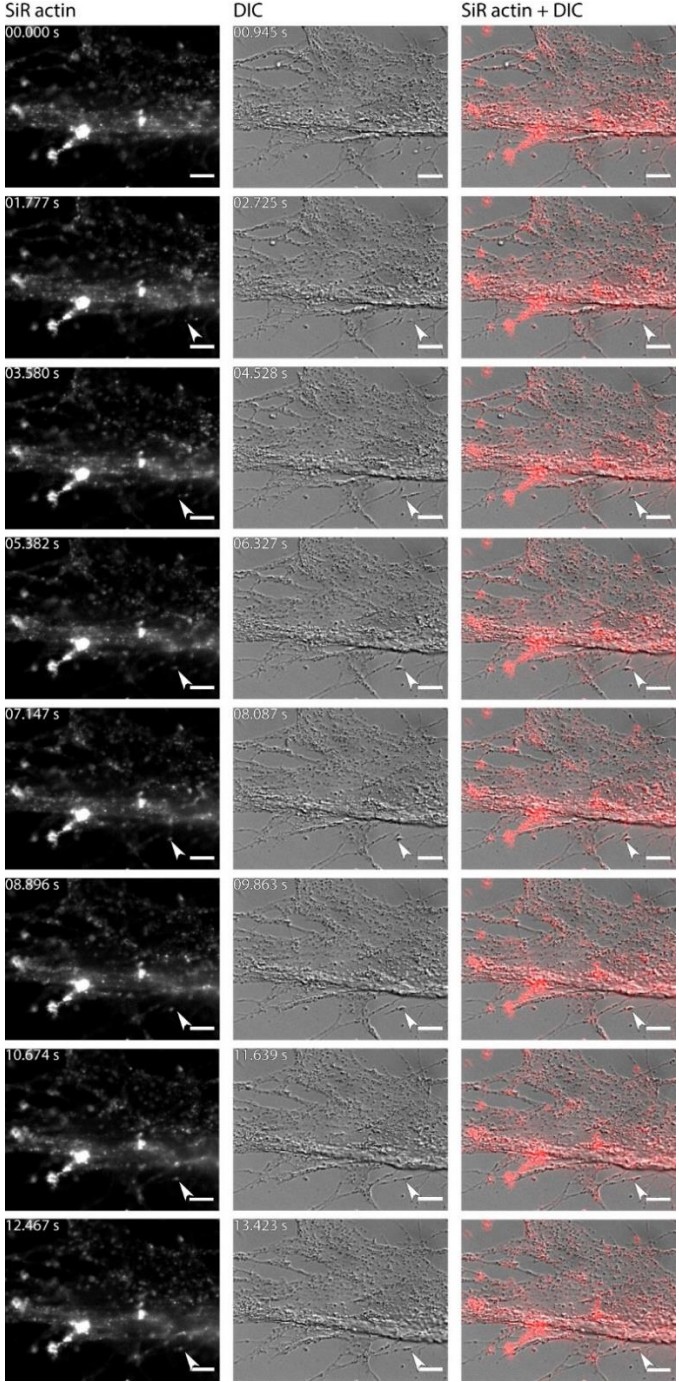

**Figure 6: Dynamics of SiR-actin-labelled granules (ALGs) in reticulopodia of *Amphistegina lessonii*.** Eight frames of time lapse. Right column: actin stained with SiR actin; middle column: DIC; right column: overlay of fluorescent and DIC channels. Arrows indicate granule in the tip of one very fine thread of forming pseudopodium. Numbers in top right corner of each image of SiR actin and DIC channel indicate time of acquisition. Conventional fluorescence images obtained with *Zeiss Axio Observer Z.1*. Scale bar 10 μm.

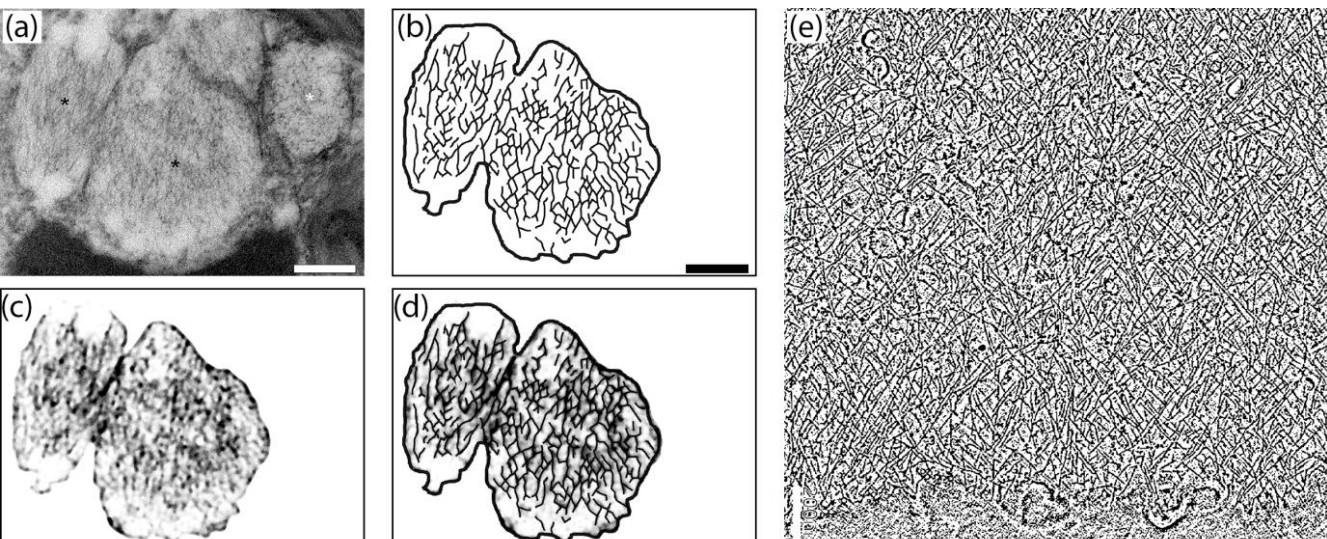

**Figure 7: Comparison of internal nanostructure of Fibrillar Vesicles (a–d) to actin meshwork (e), Scale bar 200 nm.** (a) TEM image of FV, reprinted from *Mar. Micropaleontol*, 138, LeKieffre et al., An overview of cellular ultrastructure in benthic foraminifera: New observations of rotalid species in the context of existing literature, 12-32, Copyright (2018a), with permission from Elsevier (fig.14). (b) Model of geometry of fibrillary structures inside FB based on image (a). (c) first step in drawing a model shown in (b) - fragment of image (a) with background removed and processed in FIJI software in order to make the geometry more apparent. (d) Overlay of image (c) with sketch of internal structure of FB drawn in CorelDraw. (e) structure of actin meshwork in lamellipodia based on nannotomogram reprinted from *Cell,* 171.1, Mueller et al., Load adaptation of lamellipodial actin networks, 188-200, Copyright (2017), with permission from Elsevier (fig. 4b, modified).

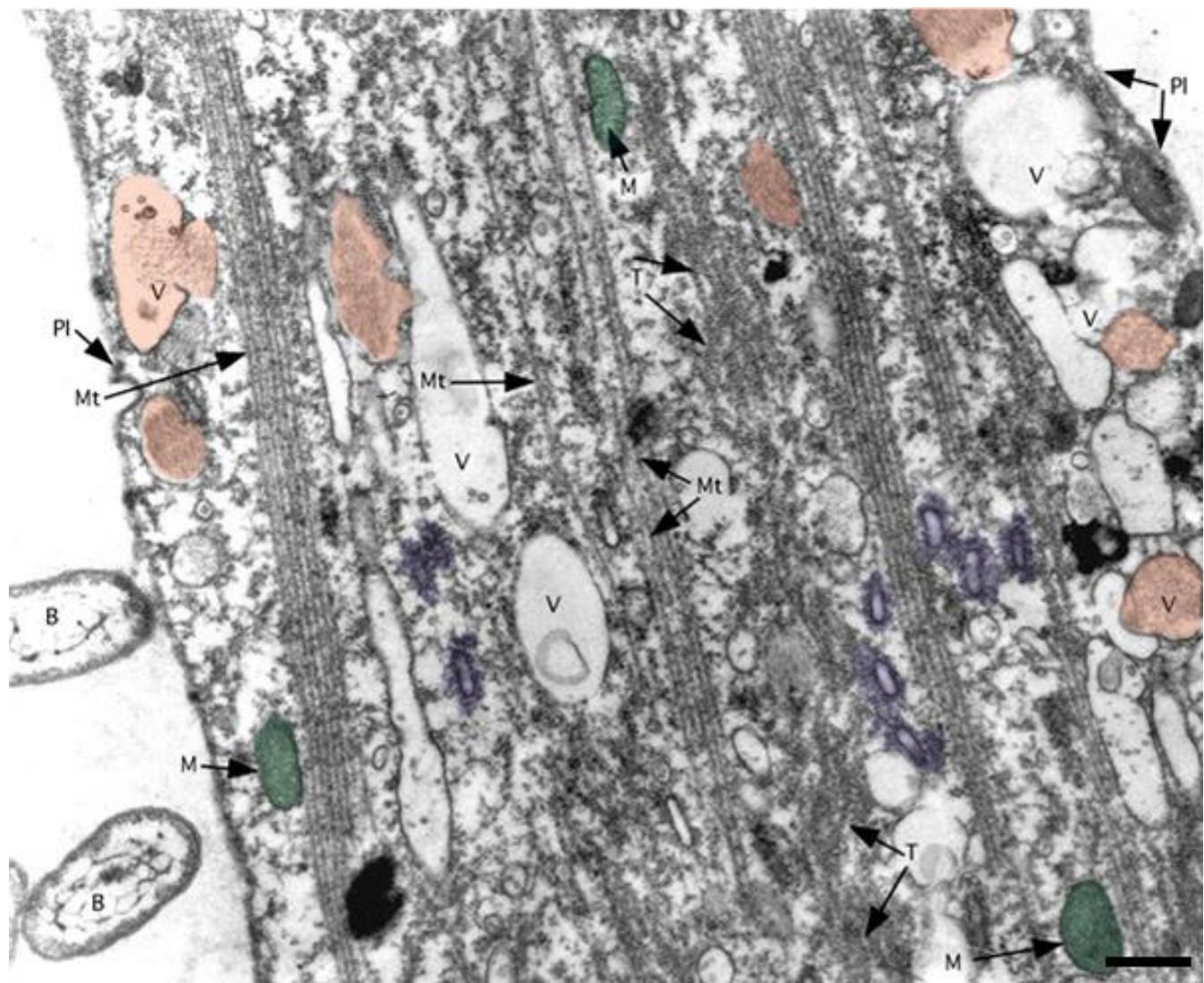

**Figure 8. TEM image of ectoplasm of** *Assilina ammonoides* (**Gronovius**) (modified from Hottinger, 2006, fig. 67 based on Creative Commons Attribution 2.5 License) Areas marked in red indicates vesicles we interpret as fibrillar vesicels. Areas marked in violet indicates what we interpret as Fuzzy coated vesicles also called Motility Organising Vesicles (MOVs). Green areas correspond to mitochondria. B: bacteria; M: mitochondria; Mt: microtubule; Pl: plasmalemma; T: tubulin paracrystals; V: vacuoles with or without fibrillar content. Scale = 500 nm.