# Peer review of "SiR-actin-labelled granules in Foraminifera: Patterns, dynamics, and hypotheses"

_Biogeosciences, 2019_

## Referee Comment (RC1) · Samuel Bowser (Referee) · 1 Jun 2019

As someone who has studied cytoskeletal structure and function for many years in various cell types, I was excited to see new information regarding the distribution of factin in foram reticulopodia! The paper represents a logical extension of research done on this topic primarily in the 1980s and 1990s. Using confocal light microscopy and a different fluorescent probe (SiR), the authors describe f-actin distribution in several pseudopodial morphotypes in a number of foram species. In this regard, the work represents a novel contribution.

The results presented are interesting, but the author's conclusions can only be considered hypothetical at this point. Of paramount importance is the correlation of fluorescence light microscopy images with electron microscopy; the simple comparisons with published photographs used here are not at all convincing. The authors should be obligated to show directly what the staining patterns correspond to ultrastructurally. (There are many straightforward ways to do this.) To be more complete, it would also be desirable to illustrate motile events (granule motion, etc.) immediately prior to fixation for electron microscopy.

Critical controls, missing from the present study, include demonstrating that the observed SiR staining patterns are not caused by the action of jasplakinolide. The authors (and sales literature) suggest that they are not, but to examine this important issue experimentally the authors should fix the cells first and then stain for f-actin using SiR and fluorescent phalloidin; equivalent patterns using two independent f-actin probes in fixed cells would be much more convincing. An important allied question is: what is the effect of unlabeled jasplakinolide on f-actin distribution and reticulopodial motility? Such information would help flesh out their study and provide important new information on the pharmacological disruption of foram cytoskeletal dynamics.

A storage form of f-actin? Because actin is \*highly\* abundant in eukaryotic cells, it would be remarkable for it to be transported as oligomers or filaments, as suggested. To make the claim believable, the authors would have to provide evidence that g-actin concentrations in reticulopods are insufficient to support localized assembly. (There is a vast literature on g-actin transport or storage forms of g-actin complexed with assembly regulatory proteins, in neurons, sperm acrosomes, etc., that the authors can consult to guide their work.)

As a final point, I question the "fit" for this study being published in Biogeoscience. It seems more suitable for a cell biology or protistology journal, where it will receive much more attention.

---

## Author Comment (AC1) · 4 Jul 2019

Thank you very much for the critical review of our contribution. We greatly appreciate the time and effort to provide us with constructive feedback. We are glad to hear that our work represents a novel and interesting contribution. Our study is a logical extension of research on F-actin in foraminiferal reticulopodia done primarily in the 1980s and 1990s.

Possibly, the most novel aspect of our research is the introduction of live actin staining in foraminifera using the SiR-actin fluorescent probe. We therefore observe and describe the organization of F-actin in action that is undisturbed by fixation. Although this

method was already partly presented in the study by Tyszka et al. (2019), our paper has a different objective focused on description of intriguing granular microstructures observed in a much smaller scale. This study is based on observations of various pseudopodial structures identified in very different taxa.

Besides many profits, a new method often comes with additional problems and questions. The fact that we demonstrate granularity as a main feature of the actin cytoskeleton in foraminifera is one of them. We are aware that a certain dose of skepticism is needed when such an unexpected finding appears using a novel technique. However, we have observed the same consistent pattern across a wide range of species using various microscopes, including Leica, Olympus, and Zeiss in different laboratories.

Referee's comment: The results presented are interesting, but the author's conclusions can only be considered hypothetical at this point.

Response: We agree that our conclusions can only be considered hypothetical at this point. This is partly expressed in the original test, including conclusions. At this point, we can only present staining patterns and propose hypotheses to be tested by future experiments. Our interpretation regarding the function of the observed Actin Labelled Granules is indeed largely hypothetical at this stage.

Of paramount importance is the correlation of fluorescence light microscopy images with electron microscopy; the simple comparisons with published photographs used here are not at all convincing. The authors should be obligated to show directly what the staining patterns correspond to ultrastructurally. (There are many straightforward ways to do this.) To be more complete, it would also be desirable to illustrate motile events (granule motion, etc.) immediately prior to fixation for electron microscopy.

Response: We fully agree that testing the different scenarios and conclusively settle important questions on ultrastructural analogs of the observed staining pattern in Foraminifera is necessary and requires additional detailed studies, ideally using a TEM-fluorescence correlation microscopy. This is probably the most sophisticated and timeconsuming type of experiments that needs to be carried out. Our intention is to run such correlative studies presenting directly what the staining patterns correspond to ultrastructurally. This is an excellent idea for a new collaborative project we would like to apply for. We should mention that this idea was already expressed in our conclusions, i.e. "According to our presented hypothesis, most of ALGs correspond to fibrillar vesicles (see LeKieffre et al., 2018a; Goldstein and Richardson, 2018) and/or elliptical fuzzy-coated vesicles (Travis and Bowser, 1991). This is still a working hypothesis that should be verified by correlative TEM-fluorescence methods."

We do agree that it would be most desirable to document dynamics of granules immediately prior to fixation for TEM. This would be the best experimental scenario. However, it might be reasonable to avoid standard fixatives that tend to alter actin organization during fixation. We would like to test different fixation methods. Possibly, the most optimal would be to apply a high pressure freezing (e.g., cryfixation in propane) to avoid preparation artifacts. Our guess is that documentation of all experiments would need another extensive and well-illustrated publication.

Referee's comment: Critical controls, missing from the present study, include demonstrating that the observed SiR staining patterns are not caused by the action of jasplakinolide. The authors (and sales literature) suggest that they are not, but to examine this important issue experimentally the authors should fix the cells first and then stain for f-actin using SiR and fluorescent phalloidin; equivalent patterns using two independent f-actin probes in fixed cells would be much more convincing. An important allied question is: what is the effect of unlabeled jasplakinolide on f-actin distribution and reticulopodial motility? Such information would help flesh out their study and provide important new information on the pharmacological disruption of foram cytoskeletal dynamics.

Response: We highly acknowledge all recommendations. We are currently planning additional experiments to address most of these points. At this stage we present results of replicated experiments conducted over last three years. Our intention is to identify

the problem, then to propose and discuss all working hypotheses. All our experiments indicate necessity of further extensive and collaborative studies.

Referring to the comment on the action of jasplakinolide, we compare stained and unstained (control) individuals. We did not observe any changes in the overall reticulopodial morphology nor in the dynamics after staining with SiR-actin (jasplakinolide-based probe). We have run replicated experiments indicating that SiR-actin (incl. jasplakinolide) does not disturb pseudopodial dynamics associated with chamber formation. Neither chamber morphogenesis nor biomineralization is modified. Our methodology follows staining technique described in Nature Chemistry or Nature Methods (Lukinavičius et al. 2013; 2014).

We would also like to stress that small dynamic objects stained with SiR-actin typically overlap with a subpopulation of well-defined granules visible in differential interference contrast (DIC) or in bright field images (see figs 1, 2, 5, 6 in the paper being reviewed). It seems to be clear that this type of granularity is immanent to the foraminiferal pseudopodial system. We don't think jasplakinolide induces formation of new granules. However, according to Melak et al. (2017), it is likely that untagged jasplakinolide induces F-actin assembly. Melak et al. (2017 on p. 527) suggest that "caution must be taken in live-cell imaging as SiR-actin might cause F-actin stabilization or induce actin polymerization owing to its structural similarities to Jasplakinolide", and later propose that "further studies are therefore needed to fully assess the advantages and possible limitations of SiR-actin over more established actin probes." We would like to take these points into account in future experiments and projects.

We would like to change the title of the paper to avoid interpretative connotations. We propose to change a title of our paper (under review) to "Actin-labelled granules in Foraminifera: Setting hypotheses based on SiR-actin live experiments". Furthermore, we propose to modify the interpretative part of the text that should describe and discuss all working hypotheses. Our intention with this manuscript was to present possible hypotheses based on our and published data and to propose the best research strategy

for designing future experiments.

Here we propose alternative explanations for the observed Actin Labeled Granules (see Fig. 1 in supplement to this response; to be proposed as a new figure in our paper). We describe three possible scenarios in which SiR-actin specifically labels F-actin within structures that have a granular appearance (Fig. 1A-C in supplement), and present possible artifacts caused by staining foraminiferal reticulopodia with SiR-actin (Fig. 1D-F in supplement). The most likely scenario assumes that SiR actin labels actin filaments inside vesicles separated from the rest of the protoplasm with a lipid membrane, possibly corresponding to Fibrillar Vesicles known form TEM ultrastructure studies (Fig. 1A in supplement). This hypothesis is discussed in detail in our manuscript. The second scenario assumes that actin filaments, surrounding some membranous vesicles, are stained specifically with SiR-actin (Fig. 1B in supplement). These vesicles may contain different kinds of cargo and F-actin is assumed to play a role in endocytosis and/or in transport. Alternatively, they may represent elliptical fussy-coated vesicles described by Koonce et al. (1986), and involved in regulation of a motility of reticulopodia Travis and Bowser (1991). The third scenario is a combination of first two as it assumes that actin filaments are both inside and outside of vesicles (Fig. 1C in supplement). Three additional scenarios assume that SiR-actin does not stain functional actin filaments in Actin Labeled Granules in foraminiferal pseudopods. These scenarios include unspecific labelling of proteins (or other complex molecules) different than actin, but mimicking a similar structure, inside (Fig. 1D in supplement) or outside (Fig. 1E in) membranous vacuoles. The last scenario assumes that SiR-actin induces assemblage of actin filaments in specific regions of cytoplasm rich in G-actin (Fig. 1F in supplement) that follows comments by Melak et al. (2017). We will stress multiple scenarios in the final version of the manuscript and discus them more extensively.

As mentioned above, we will follow up on the suggestion to monitor the movement of granules prior to fixation and perform correlative light-electron microscopy to validate if

ALGs (or a subset of them) are indeed identical with Fibrillar Vesicles.

We are conducting additional control experiments, including phalloidin and SiR-actin parallel staining and monitoring of Actin Labelled Granules in living specimen treaded with inhibitors of F-actin polymerization. We expect a high degree of overlap between the signal form phalloidin and SiR-actin but not necessarily 100 % correlation due to bonding to different epitopes on the F-actin surface. On the other hand, the risk of fixation artifacts can never be discarded. Some granules/vesicles might react to fixation by fusing or dispersing. Another problem might be related to detergents used for permeabilization that might break or modify granules, as they may affect not only the cell membrane but also internal organelles, including Actin Labelled Granules, if our assumption is correct and they consist of densely packed actin filaments enveloped in lipid membrane).

We also agree that the impact of unlabeled jasplakinolide on the motility and morphology of reticolopodia and the F-actin distribution is worth testing in details. As far as we observe, labelled jasplakinolide (i.e. SiR actin) is not disturbing the movement of reticulopodia or chamber formation at all. Either with or without labelling, chamber formation works the same way. The density of granules (seen in transmitted light), as well as their speed are comparable. It is worth mentioning that it was shown for animal cells that the cytotoxic effect of labelled jasplakinolide is much lower than unlabelled (Lukinavičius et al. 2014). More important is the deteriorating impact of laser light, especially during longer experiments (overnight time lapses).

Referee's comment: A storage form of f-actin? Because actin is *highly* abundant in eukaryotic cells, it would be remarkable for it to be transported as oligomers or filaments, as suggested. To make the claim believable, the authors would have to provide evidence that g-actin concentrations in reticulopods are insufficient to support localized assembly. (There is a vast literature on g-actin transport or storage forms of g-actin complexed with assembly regulatory proteins, in neurons, sperm acrosomes, etc., that the authors can consult to guide their work.)

Response: We are aware that such a mode of transport would be remarkable in the eukaryotic system. However, this mode might represent an analog to tubulin paracrystals described in foraminifera by the reviewer and collaborators. With regard to the presence of G-actin in reticulopodia, we plan to perform experiments in near future to measure the G-actin/F-actin ratio in reticulopodia and to estimate if the G-actin content in reticulopodia is enough to explain the presence of the observed F-actin structures. We agree that G-actin is very abundant in eukaryotic cells, hence the proposed system of transporting prefabricated actin filament seems to be unusual. Foraminiferal cells, however, differ from most of other eukaryotic cells in their size and ability to rapidly extend pseudopods. The abundance of G-actin seems to be restricted to endoplasm. Reticulopodia, and especially their distal parts may differ in that regards, as they do not contain ribosomes (Bowser and Travis, 1991). Actin among other structural components must be somehow transported to those places, simple diffusion may not be sufficient. It was mentioned in the referee's comment that some cells, such as neurons have systems of G-actin transport. Even though there are many analogies between axon growth and reticulopodia extension in foraminifera, and they share many physiological mechanisms, there are also some significant differences. The most prominent difference between neurons and foraminifera is the time scale of morphogenetic processes. Travis and Bowser (1991) state that foraminiferans extends pseudopods at speeds in excess of 1 $\mu$m/s [...]. In contrast, neurite outgrowth from neurons cultured at 37°C (albeit only a superficially similar process) occurs at approximately 10 $\mu$m/h. Similar growth rates of neurites are found in several other studies as well, e. g. for Xenopus a growth rate of 54.6+1.22 $\mu$m h-1 has been reported (Konopacki et al. 2016). Moreover, there is growing evidence that even in neurons there are some systems of transport and turnover of prefabricated elements including lipid membranes and numerous receptors (Vitriol and Zheng, 2012).

If transportation of actin cytoskeleton components from the endoplasm to reticulopodia in foraminifera occurs as proposed in our hypothetical model (in form of discrete portions most likely separated from the rest of the protoplasm with lipid membrane), it

can easily be controlled by the foraminiferal cell. This may explain coordinated and directed movements displayed typically by pseudopodia. If our hypothesis is correct then the membrane covering the ALGs may also serve an important regulatory function requiring the co-transport of various additional membrane proteins (such is receptors or proteins involved in membrane fusion or mechanical properties).

Taking into account published data (Bowser et al. 1988), one may assume that actin and tubulin are two complementary parts of the system responsible for morphogenesis and support of the form of reticulopodia where they possibly serve two opposite functions: tubulin provides stiffness and actin is mainly responsible for adhesion, elasticity and the dynamic aspects of reticulopodia.

We cannot rule out the possibility that G-actin is transported within special vesicles and that SiR-actin induces the assemblage of actin filaments within them (Fig. 1F in supplement). What we call Actin Labeled Granules may actually be a transportation vesicle of a concentrated solution of G-actin. In that case jasplakinolide would just initiate assembly (polymerization) of F-actin. If this were true, it would definately be an interesting physiological property of Foraminifera (Fig. 1F in supplement). However, this hypothesis seems to be less likely than the hypothesis that ALGs primarily serve as a transportation and storage vehicle of prefabricated F-actin, because Fibrillar Vesicles (FVs) known from TEM images may correspond to ALGs. FVs have a similar size and their internal structure is compatible with this hypothesis. An alternative hypothesis assumes that ALGs contain both prefabricated F-actin (at least oligomers) and some pool of G-actin.

In conclusion, all these presented hypotheses should be verified and tested by further extensive studies. We hope that our submitted and discussed contribution is a good motivation to carry on such complex studies.

Referee's comment: As a final point, I question the "fit" for this study being published in Biogeoscience. It seems more suitable for a cell biology or protistology journal, where

it will receive much more attention.

Response: This true that this paper seems more suitable for a cell biology or protistology journal. It might receive much more attention. However, Foraminifera is a model group of organisms critical in Earth sciences. We would like to present our results in Biogeosciences because this journal perfectly links "bio-" with "geo-sciences". This journal has published many studies investigating recent foraminifera or different physiological processes such as mechanisms of biomineralization. F-actin is indeed involved in biomineralization in Foraminifera, we have observed dynamic Actin Labeled Granules within globopodium and lamellipodium during chamber formation and biomineralization. We would like to stress this fact in the final version of our manuscript. Moreover, presence of Actin Labeled Granules is very unusual feature of this taxon may have a great evolutionary significance. We have to admit that most cell biology journals neglect marine protists, being focused on model organisms, such as Drosophila melanogaster, Saccharomyces cerevisiae, Arabidopsis thaliana, Zea mays or mammalian cells. Just giving a single example, Nature Cell Biology has never referred to Foraminifera in any published paper. We believe that Creative Commons License offered by BG offers the best strategy to cross the BIO/GEO "demarcation line".

Thank you very much for all valuable suggestions and constructive comments.

Bibliography:

Konopacki, F. A., Wong, H.H., Dwivedy, A., Bellon, A., Blower, M. D., and Holt, C. H. ESCRT-II controls retinal axon growth by regulating DCC receptor levels and local protein synthesis. Open biology, 6.4, 150218, https://doi.org/10.1098/rsob.150218, 2016.

Koonce, M. P., Euteneuer, U., McDonald, K. L., Menzel, D., and Shliwa, M. Cytoskeletal architecture and motility in a giant freshwater amoeba, Reticulomyxa. Cell Motil. Cytoskeleton, 6.5, 521-533, https://doi.org/10.1002/cm.970060511, 1986.

Lukinavičius, G., Umezawa, K., Olivier, N., Honigmann, A., Yang, G., Plass, T., Mueller,

[Figure]

V., Reymond, L., Corrêa I. R. Jr., Luo, Z., Schultz, C., Lemke E. A., Hppenstall, P., Eggeling, C., Manley, S., and Johnsson, K. A near-infrared fluorophore for live-cell super-resolution microscopy of cellular proteins. Nature Chemistry, 5(2), 132, https://doi.org/10.1038/nchem.1546, 2013.

Lukinavičius, G., Reymond, L., D'este, E., Masharina, A., Göttfert, F., Ta, H., Güther, A., Fournier, M., Rizzo, S., Waldmann, H., Blaukopf, C., Sommer, C., Gerlich, D. W., Arndt, H, Hell, S. W., and Johnsson, K. (2014). Fluorogenic probes for live-cell imaging of the cytoskeleton. Nature Methods, 11(7), 731, https://doi.org/10.1038/nmeth.2972, 2014. Melak, M., Plessner, M., and Grosse, R. Actin visualization at a glance. J Cell Sci, 130(3), 525-530, https://doi:org/10.1242/jcs.189068, 2017.

Vitriol, E. A., and James Q. Zheng. Growth cone travel in space and time: the cellular ensemble of cytoskeleton, adhesion, and membrane. Neuron, 73.6 1068-1081, https://doi.org/10.1016/j.neuron.2012.03.005 2012.

Travis, J. L., and Bowser, S. S.: The motility of Foraminifera. In: Biology of Foraminifera, edited by: Lee J. J., Anderson O. R., Academic Press (Harcourt Brace Jovanvich), London, San Diego, New York, Boston, Sydney, Tokyo, Toronto, 90-155, 1991.

Travis, J. L., Bowser, S. S., Calvin, J. G., and Lee, J. J. (1988). Pseudopodial tension in Amphisorus hemprichii, a giant Red Sea foraminifer. In: Cell Dynamics, edited by: Tazawa M. Springer, Vienna, 64-71, 1988.

Please also note the supplement to this comment:
https://www.biogeosciences-discuss.net/bg-2019-182/bg-2019-182-AC1-supplement.pdf

**Supplement:**

[Figure]

Fig. **1.** Graph showing 6 possible scenarios of staining foraminiferal cells with SiR-actin. **A-C** shows hypothetical scenarios of specific staining: **A** illustrates a scenario in which SiR-actin stains actin filaments in the interior of vesicles (e.g. Fibrillar Vesicles); **B** SiR-actin labels actin filaments surrounding vesicles possibly related to movement of these vesicles (e.g. facilitating endocytosis or corresponding to elliptical fussy-coated vesicles). **C** assumes that actin filaments (stained with SiR-actin) are both inside and outside vesicles. **D-E** illustrate potential staining artifacts. In those scenarios SiR-actin does not label physiologically active actin filaments in granules in foraminifera (ipso facto observed Actin Labelled Granules consist artifacts), incl. unspecific labelling of some proteins inside (**D**) or outside (**E**) of vesicular structures, or (**F**) SiR-actin induction of F-actin polymerisation.

---

## Referee Comment (RC2) · Takashi Toyofuku (Referee) · 14 Jul 2019

General comments

This study used a fluorescent probe "SiR-actin" that specifically stains actin filament. The authors describe whether or not it is actin with various potential pieces of evidence. Further, the authors recorded the behaviour of stained materials. In particular, they describe the movement of particles that are present on the pseudopodia. The particles were transported rapidly on the pseudopodia. The authors are assuming that it is a "packet of actin filament". It is hypothesized that it is one of the causes for the pseudopodia of foraminifers to be rapidly extendable and retractable. I like this series

of observations and estimations from a general point of view. The content is complementary to Tyszka et al. (2019), which was previously published on ProNAS.

The results are presented in beautiful photomicrographs and are ambitious scientific manuscripts with new suggestions. I'm positive about this manuscript, but there are some points that I would like authors to improve for publication.

They suggest that what is stained in SiR-actin were the membranous surfaces of pseudopodial structures, linear or ring-like structures, and small but strongly labelled granular structures. Then, they defined these small but strongly labelled granular structures as actin-labelled granules (ALGs). The behaviour of ALGs is partially documented.

Furthermore, since mitochondria are known to distribute on the pseudopodia, they distinguish ALGs from mitochondria by Mitotracker green. Mitochondria should be indicated by Mitotracker green and not by SiR-actin. Then, the authors deny the possibility that ALGs are mitochondria. That is the reason why particles stained with SiR-actin present on the pseudopodia are not mitochondria but actin.

In interpretation and argument, it is claimed that the materials stained with SiR-actin are actin, this would be over-interpretation. Since this point is the limit of the fluorescent staining method. Argue the certainty from the comparison of the current results with the previous studies, TEM, and the fact that the distribution with mitochondria does not overlap. I would like you to discuss this point clearly and collectively in section 4.1. The possibility of the existence of the structure observed in the TEM of the previous studies should be included in this paragraph. If the authors will not discuss reliability and robustness in 4.1, readers cannot consider the following argument.

Description of the methodology that can be reproduced experimentally is essential for the scientific paper. Basic information such as what you observed with the filter set is missing in this study. The authors need to improve the writing of methodology. This point is the most unacceptable problem in this manuscript.

It is overestimated and misleading to call those stained with SiR-actin as actin-labelled granules (ALGs).

Why not describe the structures stained with SiR-actin other than granule-like structures? I think it's as interesting as granules, so be sure to discuss it.

Discussions about evolutionary analysis (P12 L1) cannot be argued without actin live imaging in other taxa.

Questions and Comments

P1. L26 Correlative fluorescent.... It is not done in this study. Is the description necessary in the abstract?

P3. L20 Describe the exact number of used species in this study.

P4. L14 How long? How about food material? What will be labelled by calcein-AM with calcium-free seawater?

P4. L16 Categorize by purpose, not the institute.

P4. L16 Refer to Ohno, Y., et al. "Cytological Observations of the Large Symbiotic Foraminifer Amphisorus kudakajimensis Using Calcein Acetoxymethyl Ester (vol 11, e0165844, 2016)." PLOS ONE 12.4 (2017).

P4. L17 remove ")"

P4. L19 Indicate the excitation and emission wavelengths of all probes.

P4. L20 " it is not possible to use this probe to label F-actin within the endoplasm" The authors show the SiR-actin fluorescent in the cell (Fig. 4). Explain exactly. In fact, it is difficult to distinguish between signals and autofluorescent from chlorophyll.

P4. L22 The descriptions are not enough to reproduce the experiment. How did authors decide experimental/observation settings (e.g. excitations/emissions and exposure times) of each probe? The settings of conventional optical observation should

be indicated, too. Were there some negative controls? What was the frequency of time-lapse imaging?

P4. L22 Indicate the setting of fluorescent cubes.

P4. L30 " tripled in some images." Did such images used for analyses? What criteria do you use to decide whether to use images or not? How many shooting cases are there?

P5. L5 Be sure to indicate whether each image is by laser confocal or by ApoTome.

P5. L6 Rather than being confirmed to be actin, it is used in the sense of particles stained with SiR-actin. Is it not misleading?

P5. L8 Put some description of DIC observation in Materials and Method.

P5. L11 This text contradicts that the cytoplasm has symbiotic algae and is unobservable.

P5. L12 Had authors mixed and discussed the results of different species? Are there variations among species?

P6. L1 Isn't "SiR-"actin-labelled granules?

P6. L3 Show the variation of dynamics. Can you make a summarized table?

P6. L6 It is a good approach.

P6. L15 Although it is difficult to measure the dynamics of the granule, the authors observe the movement of the fluorescently labelled granules that were seen along the pseudopod. This makes it possible to observe the movement of granules by limiting into a one-dimensional movement. For example, if time is plotted on the horizontal axis and the coordinates in the pseudopodia on the vertical axis, can it be possible to illustrate temporal changes in granule's position.

P6. L19 How did you calculate the rate of granule movement? Did you show the

measurement method in the materials and methods?

P6. L22 Indicate the dynamics of other types of a granule.

P7. L5 FLAKOWSKI et al can be found in the reference list. I guess the authors refer to the study here. Discuss the relationship between foraminiferal actin variability and phylogeny.

P7. L18 "Such effects have not been reported." Did you compare the results with the population of negative controls?

P8. L10 Please reconsider the subtitles. It does not match the content.

P8. L16 This paragraph plays the role of the just introduction of following chapters not a discussion of results. Reconstruct the chapter structure. e.g. The chapter number 4.4 should be 4.3.1.

P10. L9 4.5.1 is numbered twice.

P10. L13 "representstructurally" insert space.

P21. L3 What does the cloudy distribution of SiR-actin around endosymbiont? No signal was detected in the same region by calcein-AM.

---

## Author Comment (AC2) · 1 Aug 2019

Dear Dr. Toyofuku,

we would like to thank you for all comments and suggestions, which help us significantly improve quality of our manuscript. Below the entire review is pasted in italics and our responses are given in regular font and follow individual comments.

**General comments**

*This study used a fluorescent probe "SiR-actin" that specifically stains actin filament. The authors describe whether or not it is actin with various potential pieces of evidence. Further, the authors recorded the behaviour of stained materials. In particular, they describe the movement of particles that are present on the pseudopodia. The particles were transported rapidly on the pseudopodia. The authors are assuming that it is a "packet of actin filament". It is hypothesized that it is one of the causes for the pseudopodia of foraminifers to be rapidly extendable and retractable. I like this series of observations and estimations from a general point of view. The content is complementary to Tyszka et al. (2019), which was previously published on ProNAS.*

*The results are presented in beautiful photomicrographs and are ambitious scientific manuscripts with new suggestions. I'm positive about this manuscript, but there are some points that I would like authors to improve for publication.*

**Re**: Thank you very much for detailed review of our manuscript. We highly acknowledge your encouraging comments on the series of our observations on SiR-actin staining experiments. We especially appreciate constructive criticism focused on methodological aspects of our studies. We are confident all the remarks and suggested changes will help us significantly improve the quality of our manuscript. We further deeply appreciate comments regarding quality of presented images.

*They suggest that what is stained in SiR-actin were the membranous surfaces of pseudopodial structures, linear or ring-like structures, and small but strongly labelled granular structures. Then, they defined these small but strongly labelled granular structures as actin-labelled granules (ALGs). The behaviour of ALGs is partially documented.*

*Furthermore, since mitochondria are known to distribute on the pseudopodia, they distinguish ALGs from mitochondria by Mitotracker green. Mitochondria should be indicated by Mitotracker green and not by SiR-actin. Then, the authors deny the possibility that ALGs are mitochondria. That is the reason why particles stained with SiR-actin present on the pseudopodia are not mitochondria but actin.*

*In interpretation and argument, it is claimed that the materials stained with SiR-actin are actin, this would be over-interpretation. Since this point is the limit of the fluorescent staining method. Argue the certainty from the comparison of the current results with the previous studies, TEM, and the fact that the distribution with mitochondria does not overlap. I would like you to discuss this point clearly and collectively in section 4.1. The possibility of the existence of the structure observed in the TEM of the previous studies should be included in this paragraph. If the authors will not discuss reliability and robustness in 4.1, readers cannot consider the following argument.*

**Re**: We would like to thank Dr. Toyofuku for this comment. We agree that other possible scenarios explaining observed staining pattern must be explicitly described in the discussion and interpretation in the final version of manuscript. To avoid risk of over-interpretation, we will follow your comments and propose to re-write the discussion section in the final version of the manuscript. We will present all possible scenarios. As pointed out in earlier comments by Dr. Samuel Bowser, we have already tried to address this issue in our response. We also add a figure illustrating different possible scenarios, explaining patterns of SiR-actin staining that we observe to the final version of supplementary materials (see Fig. 1 – this figure was previously submitted as a supplement to Dr. Bowser's comment). To address both referee's concerns, we would like to propose adding to the section "4.1 Assessment of unspecific fluorescent labelling risk" this following paragraph:

"As the granular pattern of SiR-actin staining is unusual compared to other Eukaryotes, it requires discussion of possible scenarios explaining this pattern (see Fig. 1 [in this response; proposed as a new figure in our paper or supplement]). Three possible scenarios in which ALGs are real F-actin-containing structures that are labelled by SiR-actin probe (Fig. 1A-C), and 3 other possibilities explain observed pattern as artifacts (Fig. 1D-F). First scenario (Fig. 1A), that seems the most likely, assumes that foraminifera possess granular structures willed with densely packed actin filaments that are specifically stained with SiR-actin. These structures possibly correspond to Fibrillar Vesicles known from TEM ultrastructure studies, see below in Section 4.5.1 According to the second scenario, labelled actin filaments surround some membranous vesicles (Fig. 1B). These vesicles are possibly involved in transport and endocytosis F-actin probably plays role in those processes. Alternatively, they may correspond to elliptical fussy-coated vesicles described by Koonce et al. (1986) regulate motility of reticulopodia (see below section 4.5.2 Elliptical fuzzy-coated vesicle). Next scenario assumes that actin filaments are both inside and outside of some membrane-bound vesicles (Fig. 1C). Furthermore, the observed staining pattern may be explained as an artifact, if SiR-actin binds to some other unidentified organic molecules that are not associated with F-actin, either inside (Fig. 1D) or outside (Fig.1E) of membranous vesicles. Finely SiR-actin may induce assemblage of actin filaments in the areas rich in G-actin (Fig. 1F in supplement) that follows comments by Melak et al. (2017)."

We would like to place this new paragraph in P7.L5.

We noticed already in P7.L15-16 of our manuscript that "the risk of interference of a probe with the physiology of actin itself, it may for instance cause an artificial polymerisation of F-actin (Melak et al., 2017)". We were also aware that hypothesis assuming that ALGs are real actin-rich granules corresponding to Fibrillar Vesicles is not sole possible scenario as we described it in our conclusions (P14.L13-15) as "a working hypothesis that should be verified by correlative TEM-fluorescence methods." We agree that we should elaborate it more in the discussion and interpretation section.

[Figure]

**Fig. 1.** Graph showing 6 possible scenarios of staining foraminiferal cells with SiR-actin. A-C shows hypothetical scenarios of specific staining: A illustrates a scenario in which SiR-actin stains actin filaments in the interior of vesicles (e.g. Fibrillar Vesicles); B SiR-actin labels actin filaments surrounding vesicles possibly related to movement of these vesicles (e.g. facilitating endocytosis or corresponding to elliptical fussy-coated vesicles). C assumes that actin filaments (stained with SiR-actin) are both inside and outside vesicles. D-E illustrate potential staining artifacts. In those scenarios SiR-actin does not label physiologically active actin filaments in granules in foraminifera (ipso facto observed Actin Labelled Granules consist artifacts), incl. unspecific labelling of some proteins inside (D) or outside (E) of vesicular structures, or (F) SiR-actin induction of F-actin polymerisation. (Published in the supplement to response to referee's comment by Dr. Samuel Bowser on the 4th of July 2018).

*Description of the methodology that can be reproduced experimentally is essential for the scientific paper. Basic information such as what you observed with the filter set is missing in this study. The authors need to improve the writing of methodology. This point is the most unacceptable problem in this manuscript.*

**Re**: We especially appreciate all methodological comments, pointing out the lack of some information on hardware setups we used in our experiments. We will add all necessary information needed for reproduction of our results.

We present detailed point-by-point response to your specific comments and questions below.

**Questions and Comments**

*P1. L26 Correlative fluorescent.... It is not done in this study. Is the description necessary*

*in the abstract?*

**Re**: Indeed it may be misleading to include this in the abstract as we did not run this type of experiments in the presented study. We will leave this out from the abstract in the final of the manuscript.

*P3. L20 Describe the exact number of used species in this study.*

**Re**: We will re-write the beginning of this paragraph, so it would more clearly specify species used for this study:

"We present here results of experiments performed on 3 species of foraminifera *Amphistegina lessonii, Ammonia* sp., *Quinqueloculina* sp.. They belong to both main classes of multilocular foraminifera (first two belong to Globothalamea and the third one to Tubothalamea). We have observed similar staining pattern in other species, such as *Calcarina* sp. and *Peneroplis* sp."

*P4. L14 How long? How about food material? What will be labelled by calcein-AM with calcium-free seawater?*

**Re**: Staining with SiR-actin and Mitotracker Green or Calcein red-range AM does not require prolonged incubation. After 15-20 minutes the signal is sufficient enough to perform observations. To minimise the problems caused by autofluorescence we starved all specimens 24 hours prior the observations.

Calcein Red-Orange, AM is a cell-permeable dye that stains cytoplasm of living cells and is often used to indicate viability of the cells. As the regular calcein it is fluorescent in presence of Calcium ions, but unlike usual calcein is cell permeable. It differs in a chemical structure from the regular calcein, and its fluorescent spectrum is shifted towards longer wavelengths.

In our experiments main purpose of using CellTrace™ Calcein Red-Orange was to indicate limits of the cytoplasm and to highlight 3D structure of pseudopodia (e.g., globopodium). Live cells structures are stained with this dye even, if they are surrounded by calcium-free artificial sea water, as calcium ions are always present within living cells.

*P4. L16 Categorize by purpose, not the institute.*

**Re**: We categorised experiments in that way because in different locations we used different equipments and setups that had variable impacts on our observations. We will follow referee's comment and stress differences in hardware used and its constrains than the location itself in the final version of the manuscript. This is true that our experiments in two different labs could be partly categorized "by purpose". Most experiments and confocal observations run at the AWI were focused on chamber formation, in contrast to experiments focused on granuloreticulopodia in the lab in Kraków.

*P4. L16 Refer to Ohno, Y., et al. "Cytological Observations of the Large Symbiotic Foraminifer Amphisorus kudakajimensis Using Calcein Acetoxymethyl Ester (vol 11, e0165844, 2016)." PLOS ONE 12.4 (2017).*

**Re**: We will add this reference and stress that fluorescent dye used for staining living cytoplasm was Calcein red-orange AM in contrast to a regular calcein that exhibits green florescence. It should also be mentioned that both fluorescent probes stain same intracellular structures.

*P4. L17 remove ")"*

**Re**: Thank you for noticing. It will be corrected in the final version of manuscript, following editorial recommendations.

*P4. L19 Indicate the excitation and emission wavelengths of all probes.*

**Re**: We will summarise necessary information in a new table added to the final version of the supplementary materials (see Table 1 below):

| Fluorescent dye | Maximum of absorption | Maximum of emission |
|---|---|---|
| SiR-actin | 652 nm | 674 nm |
| MitoTracker Green | 490 nm | 516 nm |
| Calcein red-orange AM | 577 nm | 590 nm |

**Table 1.** List of fluorescent dyes used in the presented study with their fluorescent properties.

*P4. L20 " it is not possible to use this probe to label F-actin within the endoplasm" The authors show the SiR-actin fluorescent in the cell (Fig. 4). Explain exactly. In fact, it is difficult to distinguish between signals and autofluorescent from chlorophyll.*

**Re**: We will rewrite this sentence to make it more accurate: "Absorption and emission parameters of the probe are overlapping with the autofluorescence of chlorophyll from endosymbionts, thus, distinguishing between SiR-actin signal and autofluorescence emitted from the endoplasm of *Amphistegina lessonii* or other species hosting endosymbionts is difficult."

*P4. L22 The descriptions are not enough to reproduce the experiment. How did authors decide experimental/observation settings (e.g. excitations/emissions and exposure times) of each probe? The settings of conventional optical observation should be indicated, too. Were there some negative controls? What was the frequency of time-lapse imaging?*

**Re**: Exposure time for experiments performed on Zeiss Axio Observer Z.1. was optimised using trial and error method and it varied between experiments depending on magnification and binning mode. This information will be included in captions to all illustrations in the final version of manuscript. We will add a table summarising this in the final version of supplementary material (see Table. 2 above). We will also add a table summarising information on hardware setting used during confocal experiments (see Table 3 above). We performed negative controls for SiR-actin staining (Fig. 1). We will present results of these experiments in supplementary materials.

In presented time-lapses we set up minimal interval between subsequent frames. Frequency of time lapses in experiments conducted on Zeiss Axio Observer Z.1 was limited by exposure time and time needed for changes the filters and illumination mode used for particular channels. Similarly in confocal experiments performed on Leica SP5 inverted confocal microscope the frequency was limited by scanning time.

| Figure or movie number | Fluorescent dye | Exposure time | Binning mode: | Objective | Apotome setting |
|---|---|---|---|---|---|
| Fig. 1 | SiR-actin | 1 s. | 1x1 | EC Plan-Neofluar 40x/1.30 Oil M27 | off |
| Fig. 2 | SiR-actin | 1 s. | 1x1 | EC Plan-Neofluar 100x/1.30 Oil M27 | off |
| Fig. 5 | SiR-actin | 112.22 ms | 1x1 | Plan-Apochormat 20x/0.8 M27 | off |
| Fig. 6 | SiR-actin | 50 ms | 3x3 | EC Plan-Neofluar 100x/1.30 Oil M27 | off |
| Fig. S3 | SiR-actin | 282.13 ms | 3x3 | EC Plan-Neofluar 100x/1.30 Oil M27 | ON: 3 frames per image |
| Fig. S3 | MTG | 15.38 ms | 3x3 | EC Plan-Neofluar 100x/1.30 Oil M27 | ON: 3 frames per image |
| Fig. S4., S5 | SiR-actin | 185.73 mm | 3x3 | EC Plan-Neofluar 100x/1.30 Oil M27 | ON: 3 frames per image |
| Movie S1 | SiR-actin | 1 s. | 1x1 | EC Plan-Neofluar 40x/1.30 Oil M27 | off |
| Movie S3 | SiR-actin | 20 ms | 5x5 | EC Plan-Neofluar 100x/1.30 Oil M27 | off |

**Table 2.** Summary of hardware settings (exposure time and binning mode) in experiments conducted on Zeiss Axio Observer Z.1.

| Figure or movie number | Emission bandwidth (MTG) | Emission bandwidth (Calcein red-orange AM) | Emission bandwidth (SiR-actin) | Excitation Beam Splitter |
|---|---|---|---|---|
| Fig. 3 | 500nm – 533nm | N/A | 650nm - 690nm for SiR-actin | FW TD 488/561/633 |
| Fig. 4 | N/A | 596nm - 616nm | 656nm - 696nm | FW TD 488/561/633 |
| Fig. S1 | 500nm - 533nm for MTG | N/A | 650nm - 690nm for SiR-actin | FW TD 488/561/633 |
| Fig. S2 | N/A | N/A | 650nm - 690nm for SiR-actin | FW TD 488/561/633 |
| Movie S2 | 500nm – 533nm | N/A | 650nm - 690nm for SiR-actin | FW TD 488/561/633 |

**Table 3.** Summary of hardware settings in experiments conducted on Leica SP5 inverted confocal microscope.

[Figure]

**Fig. 2. Globopodia of *Amphistegina lessonii* observed using bright-field and confocal imaging.** Images in upper row represent unstained individual (control), images in lower row represent specimen stained with SiR-actin and calcein red-orange AM. The white arrows indicate autofluorescence of endosymbionts, and black arrows indicate globopodium of newly formed chamber.

*P4. L22 Indicate the setting of fluorescent cubes.*

**Re**: We would like to add this information to the final version of manuscript. The settings are summarized in Table 4 (see below):

| Fluorescent dye | excitation filter | beamsplitter | emission filter |
|---|---|---|---|
| SiR-actin | BP 640/30 | FT 660 | BP 690/50 |
| MitoTracker Green | BP 500/20 | FT 515 | BP 535/30 |

**Table. 4**. Properties of filter sets used for different fluorescent dyes for experiments performed on Zeiss Axio Observer Z.1.

*P5. L5 Be sure to indicate whether each image is by laser confocal or by ApoTome.*

**Re**: We will follow this instruction. At this stage, confocal images seem to be described clearly enough. We will add specific information to captions of images obtained by Zeiss Axio Observer Z.1 to distinguish conventional fluorescence images from optical sectionings made by ApoTome.

*P5. L6 Rather than being confirmed to be actin, it is used in the sense of particles stained with SiR-actin. Is it not misleading?*

**Re**: Our intention is to introduce a term that would be the most neutral and open for different interpretations. We chose this term because it was based on our observations (i.e. strong signal of probe targeting F-actin), as well as followed reliability of SiR-actin probe tested on other eukaryotic cells (for overview see Lukinavičius et al., 2014; Melak et al., 2017). We agree with the reviewer that at this stage

this term should not imply any specific interpretation of function and ultrastructure of granules in question. Therefore, we would like to define ALGs as fluorescent granules labelled with SiR-actin probe. This definition should precisely describe the term ALGs and avoid misleading interpretations. Nevertheless, further experiments would either prove or disprove whether ALGs represent true actin-labelled granules or just artifacts of unspecific labelling. All these possible scenarios (see Fig. 1 above) are presented and discussed in our response to referee's comment by Dr. Samuel Bowser.

*P5. L8 Put some description of DIC observation in Materials and Method.*

**Re**: We will add a sentence at the end of the paragraph: "Nomarski contrast or Differential interference contrast (DIC) is a microscopy technique utilising interferometry principle for improving contrast in transparent objects."

*P5. L11 This text contradicts that the cytoplasm has symbiotic algae and is unobservable.*

**Re**: Fig. 5. We refer in this fragment to the image that shows ALGs in endoplasm of *Quinqueloculina sp.* – the species that does not bear the symbionts. We will make it more clear in the final version of manuscript that overlapping of SiR-actin signal and chlorophyll autofluorescence regards only some species of foraminifera. However, we are aware that strong autofluorescence of food particles might occur in all foraminiferal species.

*P5. L12 Had authors mixed and discussed the results of different species? Are there variations among species?*

**Re**: We will add name of the species in which we observed ALGs within different cytoplasmic structures (at present this information is only in the caption of the figures, we agree that it is needed to clarify the text). At least in reticulopodia, there seems to be some variations between species, for example *Amphistegina lessonii* appears to have more ALGs than *Ammonia sp.* We did not performed detailed experiments to quantify differences in density of ALGs in different taxa. We would like to dedicate a separate study to analyse this important issue.

*P6. L1 Isn't "SiR-"actin-labelled granules?*

**Re**: This is similar question to referee's comment to P5. L6. We would like to address this question by showing that those granules are stained not only with SiR-actin but also with different probes targeting F-actin such as phalloidin conjugates. We plan such experiments in the next project. We agree with our referee that at this stage ALGs are in fact fluorescent granules labelled with SiR-actin probe. This definition would precisely describe the term ALGs and avoid misleading interpretations.

*P6. L3 Show the variation of dynamics. Can you make a summarized table?*

**Re**: Yes, we will include a table in the final version summarising information velocity of the movement of ALGs. We may include detailed table showing data on motility of ALGs, as well as time lapse movies on which our calculations are based.

*P6. L6 It is a good approach.*

**Re**: Thank you for that encouraging comment to our statement that "For the sake of simplicity, particular threads of granuloreticulopodia may be considered as one-dimensional structures that constrain possible directions of the movement: they can move along the thread of reticulopodia either inward or outward"

*P6. L15 Although it is difficult to measure the dynamics of the granule, the authors observe the movement of the fluorescently labelled granules that were seen along the pseudopod. This makes it possible to observe the movement of granules by limiting into a one-dimensional movement. For example, if time is plotted on the horizontal axis and the coordinates in the pseudopodia on the vertical axis, can it be possible to illustrate temporal changes in granule's position.*

**Re**: Thank you for this valuable comment. We tried several different approaches for calculating velocity of granules. Presented in the manuscript value of the velocity (9 µm/s) was calculated using manual tracking tool in Fiji software (based on ImageJ).We will add a missing description of these methods and a reference to the authors manual tracking plug-in in the final version of manuscript.

We have tested other methods of quantification of the motility of ALGs including TrackMate plug-in in Fiji software, but we encounter some difficulties using this approach. In first step this plug-in automatically finds spots (in our case ALGs) in each of the frame of the time lapse, and in second step decides, which granules on the frame n+1 correspond to granules on frame n. This software seems to cope very well with the first step in case of images of ALGs, but the second step when done automatically is very prone to errors that can cause strong bias in results. A manual approach is in the contrary very time-consuming, thus, we calculated velocities only of small portion of observed granules, hence, our results should be treated as rather conservative estimates and actual range of the speed of ALGs may be higher.

Since then we learnt that TrackMate plug-in in Fiji software allows for manual tracking of automatically annotated spots. This approach seems promising, and following referee's comment, we would like to present some results obtained with this method in the final version of manuscript. Using this approach and some new time lapses we already recorded higher velocities of ALGs than mentioned in the manuscript (at least up to 13 µm/s).

*P6. L19 How did you calculate the rate of granule movement? Did you show the measurement method in the materials and methods?*

**Re**: We did not described it materials and methods, what is shortcoming of proposed manuscript. Thank you for pointing it out. We will add necessary information in the final version of manuscript and table with date used for calculation to the final version of supplement.

*P6. L22 Indicate the dynamics of other types of a granule.*

**Re**: The dynamics of other types of granules seem to be comparable to ALGs. Nonetheless, specific observations and measurements have not been made.

*P7. L5 FLAKOWSKI et al can be found in the reference list. I guess the authors refer to the study here. Discuss the relationship between foraminiferal actin variability and phylogeny.*

**Re**: Indeed we have referred to Flakowski et al in P12. L10. However, we made a typographic error (we rote Falkowski et al. instead of Flakowski et al.). We will correct it in the final version of manuscript. In line P12. L8- L17 we discuss evolution of actin-encoding genes in foraminifera and its hypothetical

impact on foraminiferal physiology (especially intracellular transport). In line P12. L10-11 we cited *Flakowski et al* (2005) who presented molecular data indicating duplication of actin genes in many species of foraminifera. As a consequence of this duplication, most foraminifera have at least two paralogs of actin-encoding genes. Hypothetically, this paralogs extend 'molecular tool kit' available to foraminiferal cells, making their physiology more flexible or more precisely controlled.

*P7. L18 "Such effects have not been reported." Did you compare the results with the population of negative controls.*

**Re**: We refer here to possibility of disruptive effect of SiR-actin on reticulopodial motility and morphology. Our statement (lack of such an effect) is based on qualitative observation of stained and unstained individuals. In those observations we did not recognise any apparent long-term differences in morphology or dynamics reticulopodia after stining. Occasionally we observed temporary retraction of pseudopods immediately after adding the staining solution to the petri dish. After 10-15 minutes incubation with SiR-actin this effect was not visible any more, and reticulopodia were spread again, closely resembling their structure and dynamics observed prior staining. There are some differences between species in strength of this reaction (*Amphistegina lessoni* being apparently less sensitive then *Ammonia sp.* or *Quinqueloculina sp.*).

*P8. L10 Please reconsider the subtitles. It does not match the content.*

**Re**: We will change the subtitle for "4.3 Main hypothesis regarding the function of SiR-actin-labelled granules".

*P8. L16 This paragraph plays the role of the just introduction of following chapters not a discussion of results. Reconstruct the chapter structure. e.g. The chapter number 4.4 should be 4.3.1.*

**Re**: Thank you for your suggestion. We will reconstruct structure of this chapter as you recommend.

*P10. L9 4.5.1 is numbered twice.*

**Re**: "4.5.1 Fibrillar system of planktic foraminifera" should be changed to "4.5.2 Fibrillar system of planktic foraminifera".

*P10. L13 "representstructurally" insert space.*

**Re**: It will be corrected. Thank you for finding this error.

*P21. L3 What does the cloudy distribution of SiR-actin around endosymbiont? No signal was detected in the same region by calcein-AM.*

**Re**: It is not clear why there is cloudy pattern of red florescence observed around endosymbionts. It may indicate some F-actin structures involved in the movement of endosymbionts within cytoplasm. Other possibility is that it is merely a consequence of technical constraints (not sufficient signal-to-noise ratio) and some optical properties of the test which can diffuse light causing some reflections and other

artifacts (similar but less prominent foggy pattern can be seen in the unstained control, see Fig. 1. in this response).

The second part of this question is still difficult to answer at present stage. It is something unexpected, but we observe it regularly for different species of Globothalamea that calcein red-orange AM stains clearly all pseudopodial structures but not the cytoplasm inside the test. At present we can only speculate that actually endo- and ectoplasm may differ in some important manner or they may be separated from each other by some membranous structure. Fluorescent dye in question is membrane-permeable only in the form of acetoxymethyl ester. Upon penetrating into the cell it is transformed in non-permeable form by enzymes that split ester bond, making it impossible to penetrate other areas of the cell enclosed by lipid membranes. That kind of additional internal compartmentalisation of foraminiferal cytoplasm may play a crucial role in physiology. This problem, although very interesting, is beyond the scope of presented study.

**References:**

Lukinavičius, G., Reymond, L., D'Este, E., Masharina A., Göttfert F., Ta H., Güther A., Fournier M., Rizzo, S., Waldmann, H., Blaukopf C., Sommer, C., Gerlich D. W., Arndt, H.-D., Hell S.W., and Johnsson K.: Fluorogenic probes for live-cell imaging of the cytoskeleton, Nat. methods, 11, 731–733, https://doi.org/10.1038/nmeth.2972, 2014.

Melak, M., Matthias, P., and Robert, G.: Actin visualization at a glance. J.Cell Sci., 130.3, 525-530, https://doi.org/10.1242/jcs.189068, 2017.

---

## Author Response (AR1)

Dear Prof. Kitazato,

We would like to express our gratitude for assessment and constructive reviews of the proposed manuscript 'Dynamics and organization of actin-labelled granules as a rapid transport mode of actin cytoskeleton components in Foraminifera' (BG-2019-182). In this document we would like to present our detailed response to reviewers' comments. The manuscript has been modified according to referees. The main changes include modification of the title of the manuscript, re-writing selected chapters, including Chapter 2 'Materials and Methods' and Chapter 4 'Interpretation and Discussion', as well as updating the results of measurement of the velocities of the granules we observed. We corrected typographic, punctuation, and stylistic errors throughout the entire manuscript. All these changes can be found in the marked-up version of the revised manuscript at the end of this document. We further made necessary changes to the supplementary materials as well. We hope all our efforts have clarified presentation of our latest scientific results.

Yours sincerely,

Jan Goleń

**Response to referees' comments**

**Response to an interactive comment by Samuel Bowser (Referee) on "Dynamics and organization of actin-labelled granules as a rapid transport mode of actin cytoskeleton components in Foraminifera" by Jan Goleń et al.**

*Referee comments are given in italics*

General response:

Dear Dr. Bowser,

Thank you very much for your critical review of our contribution. We greatly appreciate the time and effort to provide us with constructive feedback. We are glad to hear that our work represents a novel and interesting contribution as a logical extension of research on F-actin in foraminiferal reticulopodia done primarily in the 1980s and 1990s. Possibly, the most novel aspect of our research is the introduction of live actin staining in foraminifera using the SiR-actin fluorescent probe. We therefore observe and describe the organization of F-actin in action that is undisturbed by fixation. Although this method was already partly presented in the study by Tyszka et al. (2019), our paper has a different objective focused on description of intriguing granular microstructures observed in a much smaller scale. This study is based on observations of various pseudopodial structures identified in very different taxa.

Besides many profits, a new method often comes with additional problems and questions. The fact that we demonstrate granularity as a main feature of the actin cytoskeleton in foraminifera is one of them. We are aware that a certain dose of skepticism is needed when such unexpected findings appear using a novel technique. However, we have observed the same consistent pattern across a wide range of species using various microscopes, including Leica, Olympus, and Zeiss in different laboratories.

*Comment: The results presented are interesting, but the author's conclusions can only be considered hypothetical at this point.*
**Re:** We agree that our conclusions can only be considered hypothetical at this point. We modified the manuscript significantly to stress this fact out. We hope that our contribution fosters further studies testing all alternative hypotheses.

*Comment: Of paramount importance is the correlation of fluorescence light microscopy images with electron microscopy; the simple comparisons with published photographs used here are not at all convincing. The authors should be obligated to show directly what the staining patterns correspond to ultrastructurally. (There are many straightforward ways to do this.) To be more complete, it would also be desirable to illustrate motile events (granule motion, etc.) immediately prior to fixation for electron microscopy.*
**Re:** We fully agree that testing the different scenarios and conclusively settle important questions on ultrastructural analogs of the observed staining pattern in Foraminifera is necessary and requires additional detailed studies, ideally using a TEM-fluorescence correlation microscopy. This is probably the most sophisticated and time-consuming type of experiments that needs to be carried out. Our intention is to run such correlative studies that is an excellent idea for a new collaborative project we would like to apply for. We should mention that this idea was already expressed in our conclusions, i.e. "According to our presented hypothesis, most of ALGs correspond to fibrillar vesicles (see LeKieffre et al., 2018a; Goldstein and Richardson, 2018) and/or elliptical fuzzy-coated vesicles (Travis and Bowser, 1991). This is still a working hypothesis that should be verified by correlative TEM-fluorescence methods."

We do agree that it would be most desirable to document dynamics of granules immediately prior to fixation for TEM. This would be the best experimental scenario. However, it might be reasonable to avoid standard fixatives that tend to alter actin organization during fixation. We would like to test different fixation methods. Possibly, the most optimal would be to apply a high pressure freezing (e.g., cryfixation in propane) to avoid preparation artifacts. Our guess is that documentation of all replicated experiments would need another extensive and well-illustrated publication.

*Comment: Critical controls, missing from the present study, include demonstrating that the observed SiR staining patterns are not caused by the action of jasplakinolide. The authors (and sales literature) suggest that they are not, but to examine this important issue experimentally the authors should fix the cells first and then stain for f-actin using SiR and fluorescent phalloidin; equivalent patterns using two independent f-actin probes in fixed cells would be much more convincing. An important allied question is: what is the effect of unlabeled jasplakinolide on f-actin distribution and reticulopodial motility?*

*Such information would help flesh out their study and provide important new information on the pharmacological disruption of foram cytoskeletal dynamics.*

**Re:** We highly acknowledge all recommendations. We are currently planning additional experiments to address most of these points. We would like to publish them in the future. At this stage we present results of replicated experiments conducted over last three years. Our intention is to identify the problem, then to propose and discuss all working hypotheses. We made major changes to the proposed manuscript and supplementary material and include negative controls (comparison images of stained and unstained individuals of *Amphistegina lessonii*). All our experiments indicate necessity of further extensive and collaborative studies.

Referring to the comment on the action of jasplakinolide, we compare stained and unstained (control) individuals. In the corrected version of the manuscript we disused it more extensively. We did not observe any long term changes in the overall reticulopodial morphology nor in the dynamics after staining with SiR-actin (jasplakinolide-based probe). We have run replicated experiments indicating that SiR-actin (incl. jasplakinolide) does not disturb pseudopodial dynamics associated with chamber formation. Neither chamber morphogenesis nor biomineralization is modified. Our methodology follows staining technique described in Nature Chemistry or Nature Methods (Lukinavičius et al. 2013; 2014).

We would also like to stress that small dynamic objects stained with SiR-actin typically overlap with a subpopulation of well-defined granules visible in differential interference contrast (DIC) or in bright field images (see figs 1, 2, 5, 6 in the paper being reviewed). It seems to be clear that this type of granularity is immanent to the foraminiferal pseudopodial system. We don't think jasplakinolide induces formation of new granules. However, according to Melak et al. (2017), it is likely that untagged jasplakinolide induces F-actin assembly. Melak et al. (2017 on p. 527) suggest that "caution must be taken in live-cell imaging as SiR-actin might cause F-actin stabilization or induce actin polymerization owing to its structural similarities to Jasplakinolide", and later propose that "further studies are therefore needed to fully assess the advantages and possible limitations of SiR-actin over more established actin probes." We would like to take these points into account in future experiments and projects.

We would like to change the title of the paper to avoid interpretative connotations. We propose to change a title of our paper (under review) to "SiR-actin-labelled granules in Foraminifera: Pattern, dynamics, and hypotheses". Furthermore, we propose to modify the interpretative part of the text that should describe and discuss all working hypotheses. Our intention with this manuscript was to present possible hypotheses based on our and published data and to propose the best research strategy for designing future experiments.

We proposed alternative explanations for the observed Actin Labeled Granules (see Fig. S7 in the updated version of supplement ). We describe three possible scenarios in which SiR-actin specifically labels F-actin within structures that have a granular appearance (Fig. S7A-C), and present possible artifacts caused by staining foraminiferal reticulopodia with SiR-actin (Fig. S7D-F). The most likely scenario assumes that SiR actin labels actin filaments inside vesicles separated from the rest of the protoplasm with a lipid membrane, possibly corresponding to Fibrillar Vesicles known form TEM ultrastructure studies (Fig. S7A). This hypothesis is discussed in detail in our manuscript. The second scenario assumes that actin filaments,

surrounding some membranous vesicles, are stained specifically with SiR-actin (Fig. S7B). These vesicles may contain different kinds of cargo and F-actin is assumed to play a role in endocytosis and/or in transport. Alternatively, they may represent elliptical fussy-coated vesicles described by Koonce et al. (1986), and involved in regulation of a motility of reticulopodia Travis and Bowser (1991). The third scenario is a combination of first two as it assumes that actin filaments are both inside and outside of vesicles (Fig. S7C).

Three additional scenarios assume that SiR-actin does not stain functional actin filaments in Actin Labeled Granules in foraminiferal pseudopods. These scenarios include unspecific labelling of proteins (or other complex molecules) different than actin, but mimicking a similar structure, inside (Fig. S7D) or outside (Fig. S7E) membranous vacuoles. The last scenario assumes that SiR-actin induces assemblage of actin filaments in specific regions of cytoplasm rich in G-actin (Fig. S7F) that follows comments by Melak et al. (2017). We will stress multiple scenarios in the final version of the manuscript and discus them more extensively.

As mentioned above, we will follow your valuable suggestion to monitor the movement of granules prior to fixation and perform correlative light-electron microscopy to validate, whether ALGs (or a subset of them) are indeed identical with Fibrillar Vesicles.

We further paln to conduct additional control experiments, including phalloidin and SiR-actin parallel staining and monitoring of Actin Labelled Granules in living specimen treaded with inhibitors of F-actin polymerization. We expect a high degree of overlap between the signal form phalloidin and SiR-actin but not necessarily 100% correlation due to bonding to different epitopes on the F-actin surface. On the other hand, the risk of fixation artifacts can never be discarded. Some granules/vesicles might react to fixation by fusing or dispersing. Another problem might be related to detergents used for permeabilization that might break or modify granules, as they may affect not only the cell membrane but also internal organelles, including Actin Labelled Granules, if our assumption is correct and they consist of densely packed actin filaments enveloped in lipid membrane).

We also agree that the impact of unlabeled jasplakinolide on the motility and morphology of reticolopodia and the F-actin distribution is worth testing in details. As far as we observe, labelled jasplakinolide (i.e. SiR actin) is not disturbing the movement of reticulopodia or chamber formation at all. Either with or without labelling, chamber formation works the same way. The density of granules (seen in transmitted light), as well as their speed are comparable. It is worth mentioning that it was shown for animal cells that the cytotoxic effect of labelled jasplakinolide is much lower than unlabelled (Lukinavičius et al. 2014). More important is the deteriorating impact of laser light, especially during longer experiments (overnight time lapses).

*Comment: A storage form of f-actin? Because actin is \*highly\* abundant in eukaryotic cells, it would be remarkable for it to be transported as oligomers or filaments, as suggested. To make the claim believable, the authors would have to provide evidence that g-actin concentrations in reticulopods are insufficient to support localized assembly. (There is a vast literature*

*on g-actin transport or storage forms of g-actin complexed with assembly regulatory proteins, in neurons, sperm acrosomes, etc., that the authors can consult to guide their work.)*

**Re:** We are aware that such a mode of transport would be remarkable in the eukaryotic system. However, this mode might represent an analog to tubulin paracrystals. With regard to the presence of G-actin in reticulopodia, we plan to perform experiments in near future to measure the G-actin/F-actin ratio in reticulopodia and to estimate if the G-actin content in reticulopodia is enough to explain the presence of the observed F-actin structures. We agree that G-actin is very abundant in eukaryotic cells, hence the proposed system of transporting prefabricated actin filament seems to be unusual. Foraminiferal cells, however, differ from most of other eukaryotic cells in their size and ability to rapidly extend pseudopods. The abundance of G-actin may be restricted to endoplasm. Reticulopodia, and especially their distal parts probably differ in that regards, as they do not contain ribosomes (Bowser and Travis, 1991). Actin among other structural components must be somehow transported to those places, simple diffusion may not be sufficient. It was mentioned in the referee's comment that some cells, such as neurons have systems of G-actin transport. Even though there are many analogies between axon growth and reticulopodia extension in foraminifera, and they share many physiological mechanisms, there are also some significant differences. The most prominent difference between neurons and foraminifera is the time scale of morphogenetic processes. Travis and Bowser (1991) state that foraminiferans extends pseudopods at speeds in excess of 1 µm/s […]. In contrast, neurite outgrowth from neurons cultured at 37ºC (albeit only a superficially similar process) occurs at approximately 10 µm/h. Similar growth rates of neurites are found in several other studies as well, e. g. for Xenopus a growth rate of $54.6+1.22$ µm h$^{-1}$ has been reported (Konopacki et al. 2016). Moreover, there is growing evidence that even in neurons there are some systems of transport and turnover of prefabricated elements including lipid membranes and numerous receptors (Vitriol and Zheng, 2012).

If transportation of actin cytoskeleton components from the endoplasm to reticulopodia in foraminifera occurs as proposed in our hypothetical model (in form of discrete portions most likely separated from the rest of the protoplasm with lipid membrane), it can easily be controlled by the foraminiferal cell. This may explain coordinated and directed movements displayed typically by pseudopodia. If our hypothesis is correct then the membrane covering the ALGs may also serve an important regulatory function requiring the co-transport of various additional membrane proteins (such is receptors or proteins involved in membrane fusion or mechanical properties).

Taking into account published data (Bowser et al. 1988), one may assume that actin and tubulin are two complementary parts of the system responsible for morphogenesis and support of the form of reticulopodia where they possibly serve two opposite functions: tubulin provides stiffness and actin is mainly responsible for adhesion, elasticity and the dynamic aspects of reticulopodia.

We cannot rule out the possibility that G-actin is transported within special vesicles and that SiR-actin induces the assemblage of actin filaments within them (Fig. S7F). What we call Actin Labeled Granules may actually be a transportation vesicle of a concentrated solution of G-actin. In that case jasplakinolide would just initiate assembly (polymerization) of F-actin. If this were true, it would definately be an interesting physiological property of Foraminifera. However, this hypothesis seems to be less likely than the hypothesis that ALGs primarily serve as a transportation and storage vehicle of prefabricated

F-actin, because Fibrillar Vesicles (FVs) known from TEM images may correspond to ALGs. FVs have a similar size and their internal structure is compatible with this hypothesis. An alternative hypothesis assumes that ALGs contain both prefabricated F-actin (at least oligomers) and some pool of G-actin.

In conclusion, all these presented hypotheses should be verified and tested by further extensive studies. We hope that our submitted and discussed contribution is a good motivation to carry on such complex studies.

As a final point, I question the "fit" for this study being published in *Biogeoscience*. It seems more suitable for a cell biology or protistology journal, where it will receive much more attention. This true that this paper seems more suitable for a cell biology or protistology journal. It might receive more attention. However, Foraminifera is a model group of organisms critical in Earth sciences. We would like to present our results in *Biogeosciences* because this journal perfectly links "bio-" with "geo-sciences". This open access journal has published many studies investigating recent foraminifera or different physiological processes such as mechanisms of biomineralization. F-actin is indeed involved in biomineralization in Foraminifera, we have observed dynamic Actin Labeled Granules within globopodium and lamellipodium during chamber formation and biomineralization. We would like to stress this fact in the final version of our manuscript. Moreover, presence of Actin Labeled Granules is very unusual feature of this taxon may have a great evolutionary significance.

We have to admit that most cell biology journals neglect marine protists, being focused on model organisms, such as *Drosophila melanogaster*, *Saccharomyces cerevisiae*, *Arabidopsis thaliana*, *Zea mays* or various cells of selected mammalian taxa. Furthermore, we believe that Creative Commons License offered by BG offers the best strategy to cross the BIO/GEO "demarcation line".

Thank you very much for all valuable suggestions and constructive comments.

**Response to the interactive comment on "Dynamics and organization of actin-labelled granules as a rapid transport mode of actin cytoskeleton components in Foraminifera" by Dr. Takashi Toyofuku (a referee)**

15  *Referee comments are given in italics*

Dear Dr. Toyofuku,

we would like to thank you for all your valuable comments and suggestions, which help us significantly improve quality of
20  our manuscript. Below the entire review is pasted in italics and our responses are given in regular font and follow individual comments.

*General comments*

25  *Comment: This study used a fluorescent probe "SiR-actin" that specifically stains actin filament. The authors describe whether or not it is actin with various potential pieces of evidence. Further, the authors recorded the behaviour of stained materials. In particular, they describe the movement of particles that are present on the pseudopodia. The particles were transported rapidly on the pseudopodia. The authors are assuming that it is a "packet of actin filament". It is hypothesized that it is one of the causes for the pseudopodia of foraminifers to be rapidly extendable and retractable. I like this series of observations and*
30  *estimations from a general point of view. The content is complementary to Tyszka et al. (2019), which was previously published on ProNAS.*
*The results are presented in beautiful photomicrographs and are ambitious scientific manuscripts with new suggestions. I'm positive about this manuscript, but there are some points that I would like authors to improve for publication.*
**Re**: Thank you very much for detailed review of our manuscript. We highly acknowledge your encouraging comments on the
35  series of our observations on SiR-actin staining experiments. We especially appreciate constructive criticism focused on

methodological aspects of our studies. We are confident all the remarks and suggested changes will help us significantly improve the quality of our manuscript. We further deeply appreciate comments regarding quality of presented images.

*Comment: They suggest that what is stained in SiR-actin were the membranous surfaces of pseudopodial structures, linear or ring-like structures, and small but strongly labelled granular structures. Then, they defined these small but strongly labelled granular structures as actin-labelled granules (ALGs). The behaviour of ALGs is partially documented. Furthermore, since mitochondria are known to distribute on the pseudopodia, they distinguish ALGs from mitochondria by Mitotracker green. Mitochondria should be indicated by Mitotracker green and not by SiR-actin. Then, the authors deny the possibility that ALGs are mitochondria. That is the reason why particles stained with SiR-actin present on the pseudopodia are not mitochondria but actin.*

*In interpretation and argument, it is claimed that the materials stained with SiR-actin are actin, this would be over-interpretation. Since this point is the limit of the fluorescent staining method. Argue the certainty from the comparison of the current results with the previous studies, TEM, and the fact that the distribution with mitochondria does not overlap. I would like you to discuss this point clearly and collectively in section 4.1. The possibility of the existence of the structure observed in the TEM of the previous studies should be included in this paragraph. If the authors will not discuss reliability and robustness in 4.1, readers cannot consider the following argument.*

**Re**: We would like to thank for this comment. We do agree that other possible scenarios explaining observed staining pattern must be explicitly described. To avoid risk of over-interpretation, we followed your comments and propose to re-write the discussion section in the final version of the manuscript. We present all possible scenarios. As pointed out in earlier comments by Dr. Samuel Bowser, we have already tried to address this issue in our response. We also add a figure illustrating different possible scenarios, explaining patterns of SiR-actin staining that we observe to the final version of supplementary materials (see Fig. S7 – in the corrected version of supplementary materials). To address both referee's concerns, we add this following paragraph to the section **4.1 Assessment of unspecific fluorescent labelling risk**:

"As the granular pattern of SiR-actin staining is unusual compared to other eukaryotes, it requires an extensive discussion of all possible scenarios (see Fig. S7). We can see three possible scenarios in which ALGs may represent real F-actin-containing structures that are labelled by SiR-actin probe (Fig. S6A-C in Supplement), and three additional possibilities that would reveal the observed patterns as artifacts (Fig. S6D-F in Supplement). The first and most likely scenario (Fig. S6A in Supplement) assumes that foraminifera possess granular structures filled with densely packed actin filaments that are specifically stained with SiR-actin. These structures possibly correspond to Fibrillar Vesicles known from TEM ultrastructure studies (see below in Section 4.5.1). According to the second scenario, labelled actin filaments surround some membranous vesicles (Fig. 1B in Supplement). These vesicles are possibly involved in transport and endocytosis and F-actin probably plays role in those processes. Alternatively, they may correspond to elliptical fussy-coated vesicles described by Koonce et al. (1986) regulating motility of reticulopodia (see below section 4.5.2 Elliptical fuzzy-coated vesicle). The third scenario assumes that actin filaments are located both inside and outside of some membrane-bound vesicles (Fig. S6C). Alternatively, the observed staining pattern may be explained as an artifact, if SiR-actin binds to another, unidentified, organic molecule that is different

from, and not associated with F-actin, either inside (Fig. S6D in Supplement) or outside (Fig.S6E in Supplement) of membranous vesicles. Lastly, SiR-actin may induce assemblage of actin filaments in the areas rich in G-actin (Fig. S6F in Supplement) as suggested by Melak et al. (2017)."

We noticed already in P7.L15-16 of original version our manuscript that "the risk of interference of a probe with the physiology of actin itself, it may for instance cause an artificial polymerisation of F-actin (Melak et al., 2017)". We were also aware that hypothesis assuming that ALGs are real actin-rich granules corresponding to Fibrillar Vesicles is not sole possible scenario as we described it in our conclusions (P14.L13-15) as "a working hypothesis that should be verified by correlative TEM-fluorescence methods." All these aspects are elaborated more in the Interpretation and Discussion section.

*Comment: Description of the methodology that can be reproduced experimentally is essential for the scientific paper. Basic information such as what you observed with the filter set is missing in this study. The authors need to improve the writing of methodology. This point is the most unacceptable problem in this manuscript.*

**Re**: We especially appreciate all methodological comments, pointing out the lack of some information on hardware setups we used in our experiments. We will add all necessary information needed for reproduction of our results.

We present detailed point-by-point response to your specific comments and questions below.

**Questions and Comments**

*P1. L26 Correlative fluorescent.... It is not done in this study. Is the description necessary in the abstract?*

**Re**: Indeed it may be misleading to include this in the abstract as we did not run this type of experiments in the presented study. We left this out from the abstract in the final of the manuscript.

*P3. L20 Describe the exact number of used species in this study.*

**Re**: We re-written the beginning of this paragraph, so it would more clearly specify species used for this study:

'Experiments performed on 3 species of foraminifera *Amphistegina lessonii* d'Orbigny, *Ammonia* sp., *Quinqueloculina* sp. They belong to both main classes of multilocular foraminifera (first two species belong to Globothalamea and the third one to Tubothalamea). We have observed similar staining patterns in other species, such as *Calcarina sp.* and *Peneroplis sp.*'

*P4. L14 How long? How about food material? What will be labelled by calcein-AM with calcium-free seawater?*

**Re**:

We made major changes to the section 2.2 to answer these questions.

*P4. L16 Categorize by purpose, not the institute.*

**Re**: We change the categorization according to the referee's comment.

*P4. L16 Refer to Ohno, Y., et al. "Cytological Observations of the Large Symbiotic Foraminifer Amphisorus kudakajimensis Using Calcein Acetoxymethyl Ester (vol 11, e0165844, 2016)." PLOS ONE 12.4 (2017).*

**Re**: We added citation suggested by the referee and described in detail the fluorescent dye that we used.

*P4. L17 remove ")"*

**Re**: We corrected this typographic error.

*P4. L19 Indicate the excitation and emission wavelengths of all probes.*

**Re**: We summarised necessary information in a new table added to the final version of the supplementary materials (Table S1).

*P4. L20 " it is not possible to use this probe to label F-actin within the endoplasm" The authors show the SiR-actin fluorescent in the cell (Fig. 4). Explain exactly. In fact, it is difficult to distinguish between signals and autofluorescent from chlorophyll.*

**Re**: We rewrote this sentence to make it more accurate: 'Absorption and emission parameters of the probe are overlapping with the autofluorescence of chlorophyll from endosymbionts, thus, distinguishing between SiR-actin signal and autofluorescence emitted from the endoplasm of *Amphistegina lessonii* or other species hosting endosymbionts is difficult.'

*P4. L22 The descriptions are not enough to reproduce the experiment. How did authors decide experimental/observation settings (e.g. excitations/emissions and exposure times) of each probe? The settings of conventional optical observation should be indicated, too. Were there some negative controls? What was the frequency of time-lapse imaging?*

**Re**: We added necessary information on hardware settings in the Tables S2 and S3 in the supplementary materials and the following sentence to the manuscript:

'Hardware settings used for obtaining images are summarised in Tables S2 (exposure time, binning mode, objectives and Apotome mode in experiments conducted on Zeiss Axio Observer Z.1.), S3 (hardware settings in experiments conducted on Leica SP5 inverted confocal microscope), and S4 (filter sets used for different fluorescent dyes for experiments performed on Zeiss Axio Observer Z.1.) We optimised these settings using trial and error method.'

Moreover we added brief information on negative control in the manuscript '[c]omparison stained individual of *A. lessonii* to unstained control shows that SiR-actin fluorescent probe indeed stains endoplasmic structures in foraminifera  (Fig. S6 in supplement)' and the picture comparison of stained and unstained individual of *A. lessonii* has been included in the supplementary materials. We also added the information regarding the interval between frames used for recordings of the time-lapse.

*P4. L22 Indicate the setting of fluorescent cubes.*

**Re**: We added Table S4 summarising the settings of fluorescent cubes to the supplementary material and we referred to this table in the manuscript.

*P5. L5 Be sure to indicate whether each image is by laser confocal or by ApoTome.*

5   **Re**: We added specific information to captions of images obtained by Zeiss Axio Observer Z.1 to distinguish conventional fluorescence images from optical sectionings made by ApoTome.

*P5. L6 Rather than being confirmed to be actin, it is used in the sense of particles stained with SiR-actin. Is it not misleading?*

**Re**: We have change the term to SiR-actin-labelled granules to avoid confusion.

*P5. L8 Put some description of DIC observation in Materials and Method.*

**Re**: We added a sentence describing this technique at the end of the paragraph together with a new citation: "Nomarski contrast or Differential interference contrast (DIC) is a microscopy technique utilising interferometry principle for improving contrast in transparent objects (Lang, 1968)."

*P5. L11 This text contradicts that the cytoplasm has symbiotic algae and is unobservable.*

**Re**: To clarify we re-wrote this sentence so now it says: 'They can be identified in endoplasmic structures within the chambers of non-symbiont-bearing species such as *Quinqueloculina* sp., close to surfaces of internal walls of the test (Fig. 5).'

20   *P5. L12 Had authors mixed and discussed the results of different species? Are there variations among species?*

**Re**: We added names of the species in which we observed ALGs within different cytoplasmic structures to the main text of the manuscript (originally the names of the species were included only in the captions of the figures).

*P6. L1 Isn't "SiR-"actin-labelled granules?*

25   **Re**: As stated in the response to referee's comment to P5. L6, we have change the term to SiR-actin-labelled granules to avoid confusion. We would like to address this question by showing that those granules are stained not only with SiR-actin but also with different probes targeting F-actin such as phalloidin conjugates. We plan such experiments in the next project

*P6. L3 Show the variation of dynamics. Can you make a summarized table?*

30   **Re**: Yes, we included tables summarising information on velocity of the movement of ALGs and a time lapse movie to the supplementary materials.

*P6. L6 It is a good approach.*

**Re**: Thank you for that encouraging comment to our statement that "For the sake of simplicity, particular threads of granuloreticulopodia may be considered as one-dimensional structures that constrain possible directions of the movement: they can move along the thread of reticulopodia either inward or outward".

5 *P6. L15 Although it is difficult to measure the dynamics of the granule, the authors observe the movement of the fluorescently labelled granules that were seen along the pseudopod. This makes it possible to observe the movement of granules by limiting into a one-dimensional movement. For example, if time is plotted on the horizontal axis and the coordinates in the pseudopodia on the vertical axis, can it be possible to illustrate temporal changes in granule's position.*

**Re**: Since submission original version of the manuscript we applied another approach for measurement of velocities of ALGs.
10 We employed the TrackMate plug-in in Fiji software to track and calculate their velocities. We present update results with detailed description how we obtained them in the corrected version of manuscript and supplementary materials.

*P6. L19 How did you calculate the rate of granule movement? Did you show the measurement method in the materials and methods?*

15 **Re**:  In the corrected version of manuscript we added new section (2.4 Measurement of velocity of the Actin Labelled Granules) explaining measurement of velocities of ALGs.

*P6. L22 Indicate the dynamics of other types of a granule.*

**Re**: The dynamics of other types of granules seem to be comparable to ALGs. Nonetheless, specific observations and
20 measurements have not been made. In the future it is worth to compare dynamics of SiR-actin-labelled granules to dynamics of mitochondria.

*P7. L5 FLAKOWSKI et al can be found in the reference list. I guess the authors refer to the study here. Discuss the relationship between foraminiferal actin variability and phylogeny.*

25 **Re**: We added the citation in the section '4.1 Assessment of unspecific fluorescent labelling risk' as suggested by referee, as well as we corrected the typographic error in the surname of the cited author in the section  '4.6 Functional implications, evolutionary consequences, and future research prospect'.

*P7. L18 "Such effects have not been reported." Did you compare the results with the population of negative controls.*
30 **Re**: As referee pointed out, this issue needed clarification. To do that we made major changes to the last paragraph of the section '4.1 Assessment of unspecific fluorescent labelling risk'.

*P8. L10 Please reconsider the subtitles. It does not match the content.*
**Re**: We changed the subtitle for "4.3 Main hypothesis regarding the function of SiR-actin-labelled granules".

*P8. L16 This paragraph plays the role of the just introduction of following chapters not a discussion of results. Reconstruct the chapter structure. e.g. The chapter number 4.4 should be 4.3.1.*

**Re**: We reconstructed structure of this chapter as you recommend and re-numbered sections of this chapter accordingly

*P10. L9 4.5.1 is numbered twice.*

**Re**: We changed numbers of the sections in this chapter, hence this error has been corrected.

*P10. L13 "representstructurally" insert space.*

**Re**: It has been corrected in the final version. Thank you for finding this error.

*P21. L3 What does the cloudy distribution of SiR-actin around endosymbiont? No signal was detected in the same region by calcein-AM.*

**Re**: It is not clear why there is cloudy pattern of red florescence observed around endosymbionts. It may indicate some F-actin structures involved in the movement of endosymbionts within cytoplasm. Other possibility is that it is merely a consequence of technical constraints (not sufficient signal-to-noise ratio) and some optical properties of the test which can diffuse light causing some reflections and other artifacts.

The second part of this question is still difficult to answer at present stage. It is something unexpected, but we observe it regularly for different species of Globothalamea that calcein red-orange AM stains clearly all pseudopodial structures but not the cytoplasm inside the test. At present we can only speculate that actually endo- and ectoplasm may differ in some important manner or they may be separated from each other by some membranous structure. Fluorescent dye in question is membrane-permeable only in the form of acetoxymethyl ester. Upon penetrating into the cell it is transformed in non-permeable form by enzymes that split ester bond, making it impossible to penetrate other areas of the cell enclosed by lipid membranes. That kind of additional internal compartmentalisation of foraminiferal cytoplasm may play a crucial role in physiology. This problem, although very interesting, is beyond the scope of presented study.

**References:**

[revised manuscript text omitted]

SiR actin  DIC  SiR actin + DIC

[Figure]

| SiR actin | DIC | SiR actin + DIC |
|---|---|---|

[Figure]

**Figure 6: Dynamics of SiR-actin-labelled granules (ALGALGs) in reticulopodia of *Amphistegina lessonii*.** Eight frames of time lapse. Right column: actin stained with SiR actin; middle column: DIC; right column: overlay of fluorescent and DIC channels. Arrows indicate granule in the tip of one very fine thread of forming pseudopodium. Numbers in top right corner of each image of SiR actin and DIC channel indicate time of acquisition. ImagesConventional fluorescence images obtained with *Zeiss Axio Observer Z.1*. Scale bar 10 μm.

[Figure]

**Figure 7: Comparison of internal nanostructure of Fibrillar Vesicles (a–d) to actin meshwork (e), Scale bar 200 nm.** (a) TEM image of FV, reprinted from *Mar. Micropaleontol*, 138, LeKieffre et al., An overview of cellular ultrastructure in benthic foraminifera: New observations of rotalid species in the context of existing literature, 12-32, Copyright (2018a), with permission from Elsevier (fig.14). (b) Model of geometry of fibrillary structures inside FB based on image (a). (c) first step in drawing a model shown in (b) - fragment of image (a) with background removed and processed in FIJI software in order to make the geometry more apparent. (d) Overlay of image (c) with sketch of internal structure of FB drawn in CorelDraw. (e) structure of actin meshwork in lamellipodia based on nannotomogram reprinted from *Cell,* 171.1, Mueller et al., Load adaptation of lamellipodial actin networks, 188-200, Copyright (2017), with permission from Elsevier (fig. 4b, modified).

[Figure]

**Figure 8. TEM image of ectoplasm of** *Assilina ammonoides* (**Gronovius**) (modified from Hottinger, 2006, fig. 67 based on Creative Commons Attribution 2.5 License) Areas marked in red indicates vesicles we interpret as fibrillar vesicels. Areas marked in violet indicates what we interpret as  Fuzzy coated vesicles also called Motility Organizing Vesicles (MOVs). Green areas correspond to mitochondria. B: bacteria; M: mitochondria; Mt: microtubule; Pl: plasmalemma; T: tubulin paracrystals; V: vacuoles with or without fibrillar content. Scale = 500 nm.

---

## Author Response (AR2)

Dear Prof. Kitazato,

We would like to kindly thank you, Dr. Takashi Toyofuku and anonymous referee for the time and efforts put into assessment and review of our manuscript 'SiR-actin-labelled granules in Foraminifera: Patterns, dynamics, and hypotheses' (BG-2019-182). We have modified the manuscript according to referees' suggestions and corrected some typographic and stylistic errors. Below we present: (1) point-by-point response to referees' comments (2) list of changes to the manuscript; (3) a marked-up manuscript version.

Yours sincerely,

Jan Goleń

**Point-by-point response to referees' comments:**

In this section referees' comments are in italics and our response is in upright font.

**Dr. Takashi Toyofuku's comment:**

*I can evaluate the overall improvement is significant compared to the previous manuscript. In addition, it is an important improvement that the scope of this manuscript has been clarified by changing the title.*

Dear Dr. Takashi Toyofuku,

Thank you for the efforts put into evaluation of our paper. We implemented necessary changes in regard to your comments, questions and suggestions. Please see implemented changes below.

*Fig. 12 and 13 which do not exist in the text are shown (e.g. P11L26), while Fig. 8 is not quoted. Also, Fig. 7 refers only to 7e, not to Fig. 7 as a whole in the texts. Fix this problem.*

We change the quotation in the text in order to match it with the numbering of the figures. Quotation in P11L26 was changed from (Figs.12a, 13) to (Figs. 7a, 8) and quotation in P12L03 was change form (Fig. 12) to (Fig. 7).

*P4L13, L29: If calcium does not disappear from the cytoplasm, what is the reason for using calcium-free seawater? Further, the Authors should Indicate the evidence or previous studies that calcium will never disappear even in calcium-free seawater.*

We are aware that incubation of foraminifera in low-calcium sea water for 3 days significantly lowers calcium ions concertation within the cytoplasm so that it becomes undetectable with fluorescent probes as shown in Toyofuku et al., (2008). We did not incubate foraminifera for such long period in low-calcium sea water, instead we started observation as soon as reticulopodia were re-extended after transferring to low-calcium sea water ad application of fluorescent probes. The reason of employing low-calcium sea water was to minimise the risk of some artefacts. We clarified this is the corrected version of the manuscript.

*P5L3: The autofluorescence of Chlorophyll and its deliberate are not lost when fasted for 24 hours (Frontalini, F., et al. Journal of Geophysical Research: Biogeosciences, 2019, https://doi.org/10.1029/2019JG005113). It is more convincing for me that the intracellular food and symbiotic algae do not affect the observation of the pseudopodia.*

Thank you for this comment. We agree that the presence food particles and symbiotic algae in the endoplasm do not interfere with the observation of the pseudopodia, food particles newly captured by pseudopodia may give some fluorescent signal while being transported to the endoplasm. To avoid capturing this signal we starved the individuals of both symbiont-bearing and non-symbiotic species prior to observation. We rephrased the relevant part of text in order to clarify this issue.

*P9L21: "strength of this reaction" It is better to describe that "Amphistegina recovers faster" than the "strength" of influence.*

Thank you for this comment. We re-phrased the sentence in question according to you suggestion.

**Anonymous referee's comment:**

5  *I have read the revised manuscript as well as the earlier reviews and the responses by the authors, as well as other supporting materials made available to me. The authors have made substantial revisions to their manuscript to address the earlier reviews, and they have mostly succeeded. The review by Sam Bowser was the most critical, and he raised several very important points, including the need for correlative fluorescent microscopy and TEM. This is quite important because the authors are proposing actin structures in foraminiferal pseudopodia that have not been previously documented. The resolution provided by light*

10  *microscopy is insufficient to truly reach the conclusions that the authors suggest. In this revision, the authors have discussed the potential correlation of structures identified in TEM images published by others as the actin-labelled granules found in this study, but this remains highly speculative. These questions will not be resolved until the proper correlative work is completed, which the authors state that they would like to do this as a follow-up study.*

15  *The authors have revised this manuscript in such a way to address these major criticisms by discussing likely possibilities and indicating the need for additional work. As such, it raises interesting questions that should be of interest to a broader audience.*
We would like to express our gratitude to the anonymous referee for assessment of our manuscript. We are grateful for general comment as well as for two specific comments to certain phrases in the manuscript that was incorrect. We corrected the manuscript accordingly.

20  *I found a couple of other minor points:*

*p. 2: The Boltovskoy and Wright reference was published in 1976, not 2013.*
Thank you for pointing out this error. We corrected it in updated version of the manuscript.

25  *p. 4: "asexually reproduced clones" is incorrect. Because foraminifera may undergo meiosis during multiple fission, the young can be haploid and are not clones.*

We appreciate this comment. We have re-wrote this part and remove misleading or incorrect phrases.

**A list of all relevant changes made in the manuscript:**

Here we present the list of the changes in the latest version of the manuscript compered to version uploaded on 19th October 2019. Numbers of pages and lines refers to the location in the previous version of manuscript. Text marked in orange is newly added and  indicates parts of the text that we omit in the current version of the manuscript.

5      1)      We change spelling of words 'organization', 'reorganization', 'organizing', 'biomineralization', 'optimization', 'recognize' and similar forms to make spelling consistent throughout the manuscript.

     2)      We replaced word '' with 'structures' in P01L15 as it is more general term and better fits the context.

     3)      We removed word 'itself' form P01L18, as it is unnecessary.

10      4)      The year in the quotation of Boltovskoy and Wright in P2L09 is corrected form '' to '1976' as pointed out by anonymous referee.

     5)      In P03L21 we added word 'were' to make the sentence grammatically correct.

     6)      In P03L23 we removed sentence '' as it is not relevant to presented study.

15      7)      We re-wrote paragraph P04L03-L08 in order to clarify second issue pointed out by anonymous referee and to fulfil Dr. Takashi Toyofuku's suggestion from the first round of review to categorise experiments by purpose and not by institute where they were performed: 'We employed two  different methods of sample preparation for observation during experiments . (1) Searching for asexual reproduction events, we  monitored large individuals  of A. lessonii climbing the glass walls of the aquaria. The juvenile individuals (2-5 days old) were picked using a fine paint brush and transferred into a sterile imaging Petri dish (ibidi® polymer coverslip bottom) containing 2 ml of clean culture medium (up to ten individuals per dish). After one dark phase of the light:dark cycle, when individuals attached to the coverslip bottom, they were examined under a binocular looking for pseudopodial activity and chamber formation. This method was applied to run chamber formation experiments (for more details see supplement to Tyszka et al. 2019)'.

8) We re-phrased the paragraph P04L09-L15 to make clear why and how we used low-calcium sea water in our experiments and why the intracellular concertation of calcium did not decrease below level detectable with fluorescent probe: At(2) In the ING PAN, second method adult individuals of *A. lessonii, Ammonia* sp. and *Quiqueloculina* sp. were picked from the culture aquaria or bottles and cleaned with fine paint brushes under the binocular to remove algae and grains of sediment covering the specimens. Then, they were transferred to glass bottom Petri dishes previously treated with hydrochloric acid over 16 hours containing 2 ml low calcium free artificial sea water prepared as described in Bowser and Travis (2000). The reason for using low calcium artificial sea water was to avoid so called beading response (Bowser and Travis, 2000). This response causes transition of pseudopodia into droplets and may be provoked by different chemical and physical factors (Travis et al., 1983). After acclimation to the low calcium free sea water, when reticulopodia were extended and adhered to the glass bottom, specimens were stained and observed. At the AWI we conducted the observations mainly on chamber formation, while at the ING PAN, most investigations were focused on reticulopodia. Toyofuku et al., (2008) reported that incubation of foraminifera in low calcium sea water for 3 day significantly decreases level of intracellular calcium, in result the presence of intracellular calcium cannot be deducted with fluorescence probes. However, we always started our experiments immediately after reticulopodia were extended (usually within 0.5 h after the transfer to low calcium sea water) and within this time the signal did not decrease below the detectable level. The second method was employed for observations of reticulopodia'.

We did this changes to answer Dr. Takashi Toyofuku's question.

9) From P04L21 we removed phrase 'at ING PAN'-as this piece of information is not necessary for repeatability of our results.

10) In P05L01 we change word 'probe' for 'SiR-actin probe' as it is more specific.

11) In P05L03 we added some information to address the issue of chlorophyll autofluorescence pointed out by Dr. Takashi Toyofuku. The end of this paragraph says now: 'To minimise the problems caused by autofluorescence we focused primarily on pseudopods where endosymbiotic algae are not present. Chlorophyll and other substances present in food particles captured by pseudopods may also give some

fluorescent signal. For ensuring that such food particles are absent within observed pseudopods all specimens were starved for 24 hours prior the staining and observations'.

12) In P05L27 we change the position of quotation (Tinevez et al., 2017) and added new necessary quotation (Schindelin et al., 2012).

13) We re-phrased sentence in P06L01-02 to make it more clear: 'As the approximate size of the ALGs is 1 μm, we used 1 μm as a 'blob' size  parameter in the LoG detector'. In the same line in the second of the following sentences, we put word 'blob' in the quotation mark.

14) In the P06L23 we change quotation mark from double to single to make in consistent through the text.

15) We change the sentence in the P06L24 to better describe our obserevations: ' All fluorescence labelled  structures observed can be matched'.

16) In P08L20 we changed '' for 'There are'.

17) In P08L22 and P08L30 we changed spelling of word 'artefact'. In previous vision we used two different spellings of this word throughout the manuscript.

18) In P09L14 we corrected spelling of word 'interference'.

19) In P09L21-L22 we re-phrased a sentence describing the differences in the response to staining between species as suggested by Dr Takashi Toyofuku: ' *A. lessonii* is  less  prone to retraction of pseudopods and recovers faster than *Ammonia* sp. or *Quinqeloculina* sp. after applying staining solution'.

20) In P10L15-L16 we change '' for 'maximum velocity recorded by us (15.5 μm/s)' to be more precise.

21) In P10L26 we corrected the numbers of figures in the text. Now there is (Figs. 7a, 8) instead of (Figs.12a, 13).

22) In P11L29 we replaced phrase '' with 'an ability to accumulate'. It s more factually accurate.

23) Form P12L01 we removed the phrase '' to clarify this part of the text.

24) In P12L03 we change the number of the figure form (Fig. 12) to (Fig. 7).

25) In P12L14 we corrected spelling of the word 'according'.

26) In P13L02 we corrected the numbers of figures. Currently here is (Figs. 1-3, 5-6) instead of .(Figs. 1-10).

27) In P13L06 we change the number of the figure form (Fig. 12) to (Fig. 8).

28) In P16L27 we corrected numbers of pages of a quoted paper in references. It is '2669-2675' instead of '2669-675'.

29) In P17L03 we added pages number of a quoted paper (2823–2850).

30) In P19L17 we added new citations to the references list (Schindelin et al., 2012).

31) In P19L13 we added new citations to the references list (Toyofuku et al., 2008).

32) In P19L13 we added missing doi number to the citation.

33) In P24 in the caption to the Fig. 4 we added the name of the species shown (Amphistegina lessonii).

34) Throughout the manuscript we replaced double spaces with single ones.

Marked-up version of the manuscript:

[revised manuscript text omitted]